# LongGenBench: Benchmarking Long-Form Generation in Long Context LLMs

**Yuhao Wu[1], Ming Shan Hee[1], Zhiqing Hu[1] and Roy Ka-Wei Lee[1]**
[1]Singapore University of Technology and Design
{wu_yuhao,mingshan_hee,zhiqing_hu}@mymail.sutd.edu.sg
roy_lee@sutd.edu.sg

## Abstract

Current benchmarks like "*Needle-in-a-Haystack*" (*NIAH*), *Ruler*, and *Needlebench* focus on models' ability to understand long-context input sequences but fail to capture a critical dimension: the generation of high-quality long-form text. Applications such as design proposals, technical documentation, and creative writing rely on coherent, instruction-following outputs over extended sequences—a challenge that existing benchmarks do not adequately address. To fill this gap, we introduce *LongGenBench*, a novel benchmark designed to rigorously evaluate large language models' (LLMs) ability to generate long text while adhering to complex instructions. Through tasks requiring specific events or constraints within generated text, *LongGenBench* evaluates model performance across four distinct scenarios, three instruction types, and two generation-lengths (16K and 32K tokens). Our evaluation of ten state-of-the-art LLMs reveals that, despite strong results on *Ruler*, all models struggled with long text generation on *LongGenBench*, particularly as text length increased. This suggests that current LLMs are not yet equipped to meet the demands of real-world, long-form text generation. We open-source *LongGenBench* to promote comprehensive evaluation and improvement in this critical area, with code and data available at `https://github.com/mozhu621/LongGenBench`.

## 1 Introduction

Recent advances in large language models (LLMs) have dramatically enhanced their ability to process long text sequences, supporting applications that range from document summarization to creative writing. Leading models such as GPT-4 (Achiam et al., 2023), LLaMa-3.2 (Dubey et al., 2024), and Claude 2.1 (Anthropic, 2024a) manage context windows of up to 128K tokens, with the Claude 3 series (Anthropic, 2024b) handling inputs exceeding 1 million tokens. However, while much attention has been given to these models' ability to retrieve and understand long *input* text sequences, far less focus has been placed on their ability to generate coherent and high-quality long-form text *outputs*—a critical requirement for tasks such as design proposals and creative writing.

Long-form text generation is crucial for real-world applications that require detailed, well-structured narratives, such as document summarization (Kumar et al., 2024), creative writing (Hua & Wang, 2020; Hu et al., 2022), and comprehensive question answering (Stelmakh et al., 2022; Lee et al., 2023; Bai et al., 2024). Despite this importance, current benchmarks are limited in their ability to assess long-form generation, focusing instead on shorter text outputs ($\leq$ 2K tokens) (Fan et al., 2018; 2019a; Dasigi et al., 2021), making them unsuitable for tasks requiring outputs of $\geq$16K tokens (Bai et al., 2024). The challenge is further compounded by the lack of robust methods for evaluating these long sequences. The ability to follow instructions is essential for long text generation (*Reversed NIAH*[1]), just as effective information retrieval is fundamental for processing long-context inputs (*NIAH*(Kamradt, 2023)). However, current benchmarks do not adequately assess whether the

---

[1]Analogous to NIAH, which involves searching for a needle (retrieval) within a long input, the reversed NIAH entails placing a specific needle (instruction-following) at a designated position within a long output.

Table 1: Comparison of Long-Context LLM Benchmarks. For the retrieval tasks' datasets, we measure length based on the number of processing tokens, while for the generation tasks' datasets, we calculate the average number of generation words produced by LLMs. 'Long-length' indicates if LLMs to analyze or generate text that is at least 8K token.

| Type of Task | Benchmark | Type of Data | Avg Len | Long-Length |
|---|---|---|---|---|
| Retrieval | Longbench(Bai et al., 2023) | hybrid | ~8k | ✔ |
| | NIAH(Kamradt, 2023) | synthetic | Any | ✔ |
| | Ruler(Hsieh et al., 2024) | synthetic | Any | ✔ |
| Generation | ELi5(Fan et al., 2019b) | hybrid | ~0.2K | ✗ |
| | Longwrite(Bai et al., 2024) | synthetic | ~2.7K | ✗ |
| | *LongGenBench*(Ours) | synthetic | ~20K | ✔ |

generated text adheres to the specific directives of a prompt. For instance, a prompt may require incorporating specific information at a certain point in a lengthy document, but evaluations often fail to verify the model's compliance with such instructions. This oversight represents a significant shortcoming in benchmarking, particularly because performance under explicit constraints typically predicts outcomes in tasks with more implicit constraints, such as story generation or academic paper production. If a model struggles with explicit requirements, it is likely to underperform in scenarios with subtler constraints.

Manual evaluations, while thorough, are both costly and impractical at scale. Meanwhile, automated evaluations using "LLM-as-a-judge" methods (Zheng et al., 2024) often yield results that are difficult to interpret and may not align with human judgments, raising concerns about their reliability. This highlights the need for more specialized benchmarks capable of reliably assessing the quality of super-long-form text generation.

To address this gap, we present *LongGenBench*, a novel benchmark designed to evaluate the quality of super-long-form text generated by long-context LLMs. Unlike existing benchmarks that primarily test retrieval or reasoning over long inputs, *LongGenBench* focuses on the model's ability to generate content that follows complex instructions over extended sequences. Our benchmark introduces tasks that reflect real-world generation challenges, such as diary writing, menu planning, and urban design, where the text must adhere to specific constraints provided in the prompt. These tasks assess whether models can correctly incorporate specific details at designated points in the text, ensuring the generated content meets the requirements laid out in the prompt. By evaluating texts up to 32K tokens, *LongGenBench* is the first benchmark to systematically test the ability to generate instruction-compliant long-form content across extended lengths. Table 1 summarizes the different benchmarks supporting long-context retrieval and generation tasks.

The evaluation tasks are organized into four distinct scenarios: Diary Writing, Menu Design, Skyscraper Design, and Urban Planning, each with varying complexity. The scenarios involve sub-tasks such as single instance, range, and periodicity, simulating realistic constraints that a model must account for. This setup allows us to measure the model's ability to generate detailed, contextually rich outputs that satisfy a wide array of criteria.

In summary, our major contributions are as follows:

- To the best of our knowledge, this is the first study to address the challenge of super-long-form generation in long-context language models, highlighting the critical importance of generating coherent, high-quality text in extended contexts.

- We introduce *LongGenBench* , a comprehensive dataset that provides a diverse set of tasks specifically designed to evaluate the super-long-form generation capabilities of LLMs across varying token lengths (16K and 32K) and levels of text complexity.

- We perform extensive experiments on both open-source and closed-source models, revealing that despite their advanced capabilities, most models struggle significantly with super-long-form generation tasks, particularly in maintaining instruction adherence and coherence over long outputs.

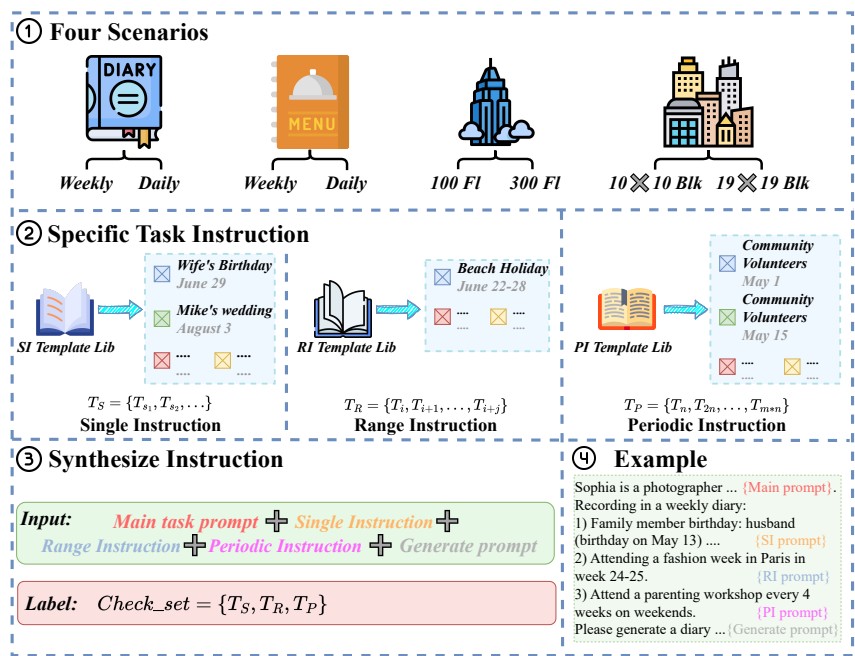

Figure 1: *LongGenBench* **Overview:** 1) **Scenario Selection:** Select from four scenarios—Diary, Menu Design, Skyscraper Design, and Urban Planning—each offered in both short and long versions to determine the main task prompt. 2) **Task Instruction:** Employ the template libraries SI (Single), RI (Range), and PI (Periodic) to generate tasks characterized by random times or locations, along with the corresponding prompts and verification sets. 3) **Instruction Synthesis:** Integrate all prompts generated in the prior step to create a comprehensive set of instructions with a final check-set. 4) **Example:** An illustration of Sophia's weekly diary task is provided as an example.

## 2 LONGGENBENCH BENCHMARK

### 2.1 TASK DEFINITION

Evaluating the quality of super-long-form generation presents a unique set of challenges due to the inherent complexity of long texts. Traditional human evaluation methods, while precise, are expensive and not scalable. Although using large language models for assessment is feasible, their lack of interpretability often hampers their utility. Thus, we focus on the "instruction-following" task in super-long-form generation, where the most must include specific details in the generated text. This task reflects real-world scenarios that require a high degree of attention to detail over extended sequences, such as technical documentation or detailed design proposals. In this study, we define a task type termed *Strictly Sequential Tasks*, which involves the sequential completion of subtasks $\mathbf{T} = \{T_1, T_2, T_3, \ldots, T_n\}$[2], where each subtask is responsible for generating a specific volume of text. For instance, an LLM might be tasked with designing a 100-floor skyscraper, specifying the content and purpose of each floor.

### 2.2 FOUR DISTINCT SCENARIO SETUPS

To comprehensively assess the long-form generation capabilities of models, we have devised four distinct task scenarios to supplement our predefined tasks, as illustrated in Figure 1 (1). These scenarios fall into two categories: Temporal (Diary Writing, Menu Design) and Spatial (Skyscraper Design, Urban Planning). Moreover, each scenario incorporates both short and long versions to assess the effectiveness of various output lengths.

These scenarios were carefully chosen to reflect both creative and technical long-form generation tasks. They offer a diverse set of challenges by including temporal tasks (e.g., Diary Writing)

---

[2]In Appendix A, there is a detailed description of the definitions of mathematical symbols.

that require maintaining consistent information over time and spatial tasks (e.g., Urban Planning) that test the model's ability to handle spatial relationships and detailed designs. These scenarios mirror real-world applications, from planning documents to creative writing, and thus provide a comprehensive evaluation of long-context LLMs. Table 2 offers comprehensive descriptions for each scenario, with each designed around a unique template to generate 100 different task instructions[3].

Table 2: Scenario task descriptions

| Category | Scenarios | Task | Task Description |
|----------|-----------|------|------------------|
| Temporal | Diary | Weekly Diary
Daily Diary | Generate entries for each week of the year
Generate entries for each day of the year |
| | Menu | Weekly Menu
Daily Menu | Plan menus for each week of the year
Plan menus for each day of the year |
| Spatial | Skyscraper Design | 100-floor Design
361-floor Design | Develop a design for a 100-floor skyscraper
Develop a design for a 300-floor skyscraper |
| | Urban Planning | 10x10 block Design
19x19 block Design | Design an urban layout on a 10x10 block grid
Design an urban layout on a 19x19 block grid |

## 2.3 SPECIFIC TASK INSTRUCTION

To enhance task control and flexibility, we have developed three distinct task settings:

- **Single Instruction** (*SI*): Injects specific information at a unique point within the generated text.
$$T_S = \{T_{s_1}, T_{s_2}, \ldots\}$$

- **Range Instruction** (*RI*): Requires the model to incorporate information within specified ranges of the text.
$$T_R = \{T_{R_i}, T_{R_{i+1}}, \ldots, T_{R_{i+j}}\}$$

- **Periodic Instruction** (*PI*): Distributes information at predefined intervals throughout the text.
$$T_P = \{T_{P_n}, T_{P_{2n}}, \ldots, T_{P_{m*n}}\}$$

- **Check Set**: Includes tasks for all three aforementioned settings.
$$Check\_set = \{T_S, T_R, T_P\}$$

For example, in the design of a 100-floor skyscraper, the *Single Instruction* may specify that the 34th floor hosts an aerial gym and the 54th floor houses a law firm. The *Range Instruction* might designate floors 1 through 9 as a comprehensive shopping mall, whereas the *Periodic Instruction* could dictate that starting from the 20th floor, every 10th floor incorporates a small aerial garden.

We utilize over 20 templates for each type of instruction, with the floors or locations being randomly assigned to ensure task diversity. These settings, applied via various templates, guarantee controlled coverage across all textual positions, thus facilitating a comprehensive and efficient evaluation, as illustrated in Figure 1 (2).

Through this approach, we generate the main task instructions $T$ and simultaneously acquire the corresponding $Check\_set$, which supports subsequent evaluations and constructs a task conducive to super-long-form generation. Subsequently, we splice the main task prompt with the specific task instructions (STI)[4] and add the generation prompt to form the final evaluation data.

## 2.4 EVALUATION METRIC

To quantitatively evaluate performance for *LongGenBench* tasks, we introduce three complementary metrics:

---

[3]Examples of Task Instructions for each scenario are provided in Appendix B.
[4]Each of our main task instructions $T$ splice 5 single instructions, 1 range instruction task, and 1 periodic instruction task.

**Main Task Completion.** This metric evaluates the extent to which all designated subtasks are accomplished. The completion rate is quantified using the following equation:

$$\text{Completion Rate (CR)} = \frac{\text{Number of Completed Subtasks}}{\text{Total Number of Subtasks}} \times 100\%$$

In this context, the numerator denotes the count of subtasks successfully executed by the model, and the denominator represents the total number of subtasks defined in the Strictly Sequential Task. For instance, does the model consistently complete a diary entry for each day without omitting any dates?

**Specific Task Instruction Completion (STIC-1).** This metric evaluates the model's adherence to specific task instructions. We calculate the completion counts for the *Single Instruction* (SI), *Range Instruction* (RI), and *Periodic Instruction* (PI). STIC-1 quantifies how well the model follows these instructions across subtasks, focusing on whether the instructions are correctly implemented. For example, in the Skyscraper Design task, if the model is instructed to place an aerial gym on the 34th floor and consistently places it on a different floor, it would receive a lower STIC-1 score.

$$\text{STIC-1} = \frac{\text{Single Instruction} + \text{Range Instruction} + \text{Periodic Instruction}}{\text{Total Number of } \textbf{Outputs} \text{ to Specific Task Instructions}}$$

**Specific Task Instruction Completion-2 (STIC-2).** STIC-2 provides a more granular assessment by measuring the overall completion of specific task instructions, including their presence and execution quality across all subtasks. In addition to adherence, it assesses whether the model consistently follows these instructions throughout the entire task. For instance, if the model periodically repeats certain elements but not at the required intervals, it would affect its STIC-2 score.

$$\text{STIC-2} = \frac{\text{Single Instruction} + \text{Range Instruction} + \text{Periodic Instruction}}{\text{Total Number of Specific Task Instructions}}$$

STIC-1 is primarily concerned with the completion rate of instructions that result in sub-scenarios, focusing on whether instructions are correctly executed. In contrast, STIC-2 assesses the overall completion of the specific instruction task, including the presence of sub-scenarios and their completion status[5].

## 2.5 EVALUATIONS PIPELINE

Our evaluation process follows a structured pipeline: First, we use a long-context LLM to complete the task instruction $T$, generating an answer $A$, which is then divided into sub-tasks as $A = \{A_1, A_2, \ldots, A_n\}$. Next, based on the specific instructions in the *check_set*, we identify the relevant sub-tasks within $A$. Finally, we evaluate each sub-task by $\text{eval}(A_i, T_i)$ to compute the final completion score, as detailed in Algorithm 1. This pipeline ensures that the evaluation is both systematic and comprehensive, assessing the model's performance across different instruction settings and levels of complexity [6]. While *LongGenBench* primarily evaluates the model's ability to follow detailed instructions, future work could expand the benchmark to include more open-ended tasks that assess creativity and logical reasoning. This would provide a broader evaluation of a model's capabilities in generating coherent, engaging, and logically sound long-form text.

## 3 EXPERIMENTS

### 3.1 EXPERIMENTAL SETUP

**Models.** We selected ten long-context large language models (LLMs), comprising eight open-source and two closed-source models. These models range in size from 7B to 72B parameters, with one featuring a Mixture of Experts (MoE) architecture. The claimed context lengths of these models vary from 32K to 128K tokens[7]. These models were selected to represent a diverse array of architectures,

---

[5]In Appendix E, we provide a detailed explanation of STIC-1 and STIC-2, along with a case study analysis.
[6]In Appendix C, there is a detailed evaluations pipeline and example.
[7]Detailed specifications of these models are provided in Appendix D.

---

**Algorithm 1** Evaluations Pipeline

---

    **Initialization:**
1:  Task instructions $\rightarrow T$
2:  Tested long context LM $\rightarrow$ model
3:  Set of Special Task Instruction for evaluation matching $\rightarrow$ Check_Set

    **Main Process:**
4:  Use Tested model to get Answer for $T \rightarrow A$
5:  $A \rightarrow \{A_1, A_2, \ldots, A_m\}$, split into subtasks
6:  empty set for storing evaluations $\rightarrow E$
7:  **for** each $T_i$ in $Check\_Set$ **do**
8:      **if** there is $A_i$ matching $T_i$ **then**
9:         eval$(A_i, T_i) \rightarrow E_i$
10:         $E \rightarrow$ Add $E_i$ to $E$
11:      **end if**
12:  **end for**
13:  $\sum E \rightarrow Score$, compute the final completion score
14:  **return** $Score$

---

covering both Mixture of Experts and standard transformer designs, as well as a range of parameter sizes. This diversity ensures a comprehensive evaluation of their ability to handle long-context tasks.

**Inference Setup.** We utilized the vLLM (Kwon et al., 2023) system, which optimizes key-value (KV) cache memory for efficient large-scale inference. This system is crucial for handling long-form generation efficiently, reducing memory overhead, and maximizing inference throughput. Inferences were performed using BFloat16 precision on $8\times$ NVIDIA A800 GPUs, employing greedy decoding to generate the outputs. This setup ensured consistency and efficiency in the inference process.

**Task Configurations.** For each scenario, we generated 800 examples at two specified lengths: 16K tokens and 32K tokens. The generation was based on designated templates for each model, ensuring task-specific relevance. The tasks were selected to reflect both creative and technical long-form generation challenges, such as diary writing, urban planning, and skyscraper design. To ensure the relevance of the generated content and prevent off-topic responses or refusals to answer, we prefixed each task input with a carefully curated answer prompt designed to guide the model's output. The tasks were specifically selected to test the models' ability to generate instruction-following long-form content in both creative and technical contexts. For example: In the *Urban Planning* task, models were tasked with generating a detailed plan for a new urban district, including descriptions of key facilities such as parks, schools, and transportation systems.

**Evaluation Metric.** We evaluated model performance using the three metrics defined in Section 2.4: *Main Task Completion*, *Specific Task Instruction Completion-1 (STIC-1)*, and *Specific Task Instruction Completion-2 (STIC-2)*. These metrics provided a comprehensive assessment of the models' ability to adhere to instructions and generate coherent long-form text.

### 3.2 MAIN RESULT

The results of the long-form text generation tasks for both Short-version (16K) and Long-version (32K) tokens are summarized in Table 3.

**Main Task Completion.** Significant disparities in performance across models primarily stem from differences in architecture and training datasets. Notably, models with varying parameter sizes, such as Llama3.1-8B-instruction (Dubey et al., 2024) (under 10 billion parameters), Qwen-72B (Yang et al., 2024) (over 20 billion parameters), and GPT-4o-mini (OpenAI, 2024a) (a closed-source model), have demonstrated superior efficacy, successfully completing most primary tasks in full. In contrast, some models struggle with these tasks, exhibiting limitations such as: 1) models responding solely to specified subtasks, neglecting others, and 2) models halting after only completing the initial task segment, despite prompts requiring full sequential subtask completion. This issue may originate from the current instructional tuning data, which could cause partial responses in complex, lengthy tasks.

Table 3: Long-form generation Performance of selected models evaluated at length from 16k and 32k. The weighted average score (wAvg) is the product of CR and STIC-2, used to represent the model's final performance at the given task length. Note that the GPT-4-32K is currently closed for use, and the longest versions that can be used are the GPT-4o and GPT-4o-mini 16K output limitation.

| Models | Claimed Length | Short-version (16K) | | | | | Long-version (32K) | | | | |
|---|---|---|---|---|---|---|---|---|---|---|---|
| | | CR | STIC-1 | STIC-2 | Len. | wAvg | CR | STIC-1 | STIC-2 | Len. | wAvg |
| *Models with 7-10B Parameters* | | | | | | | | | | | |
| Mamba-2.8B | 2K | 11.3% | 23.8% | 2.1% | 902 | 0.2% | 5.6% | 29.8% | 1.6% | 864 | 0.1% |
| FILM-7B | 32K | 36.0% | 22.4% | 9.0% | 6280 | 3.2% | 37.4% | 30.9% | 10.9% | 13775 | 4.1% |
| Mistrial-v0.2-7B | 32K | 81.8% | 25.7% | 20.4% | 7296 | 16.7% | 48.2% | 35.4% | 15.7% | 16146 | 7.6% |
| Phi-3-mini-3-8B | 128K | 22.9% | 27.6% | 5.4% | 4165 | 1.2% | 7.4% | 46.9% | 2.4% | 2613 | 0.2% |
| LLama3.1-8B | 128K | 93.5% | 23.4% | 22.0% | 8804 | 20.6% | **77.6%** | 28.9% | 20.6% | 17354 | 16.0% |
| Qwen2-7B | 128K | 60.0% | 23.3% | 13.5% | 5138 | 8.1 % | 40.0% | 31.7% | 12.6% | 9617 | 5.0% |
| LongWriter-llama3.1-8B | 128K | 46.0% | 32.6% | 14.2% | 11036 | 6.5% | 34.5% | 36.3% | 10.8% | 19925 | 3.7% |
| *Models Larger Than 20B Parameters* | | | | | | | | | | | |
| Mixtral-8x7B | 32K | 83.0% | 34.4% | 27.2% | 8113 | 22.6% | 60.5% | 36.3% | 20.3% | 15839 | 12.3% |
| Phi-3.5-8x7B | 128K | 26.9% | **46.4%** | 11.3% | 5430 | 3.0% | 7.4% | 62.9% | 6.0% | 6633 | 0.4% |
| LLama3.1-70B | 128K | 79.3% | 34.4% | 29.2% | 8055 | 23.1% | 63.1% | **43.3%** | **26.3%** | 15197 | **16.6%** |
| Qwen2-72B | 128K | 94.3% | 29.7% | 27.1% | 8013 | 25.5% | 66.2% | 34.4% | 21.7% | 19845 | 14.4% |
| *Closed-source Model* | | | | | | | | | | | |
| GPT-4o-mini | 128K | **97.0%** | 34.8% | **33.4%** | 8940 | **32.4%** | – | – | – | – | – |
| GPT-4o | 128K | 67.2% | 42.9% | 24.4% | 9055 | 15.3% | – | – | – | – | – |

Especially in GPT-4o(OpenAI, 2023), it recognizes that this task will generate a long output and only provides a few examples.

**STIC-1 and STIC-2.** The `STIC-1` metric revealed strong performance in adhering to task instructions for models like `LLama3.1-70B` and `GPT-4o-mini`, particularly in shorter sequences. However, a significant drop in `STIC-2` scores for several models indicates that maintaining instruction adherence over longer text sequences remains a challenge. This performance degradation emphasizes the need for better tuning and architectural modifications to improve long-term coherence.

A common failure mode observed across multiple models was the tendency to forget or misinterpret instructions as the sequence length increased. For example, in the *Skyscraper Design* task, some models correctly described the initial few floors but deviated from the original plan as the task progressed, particularly in the 32K token setting. This highlights the memory retention issue in long-context models, which often leads to a loss of coherence and adherence to task instructions. Examples of failures where models struggled to follow instructions are provided in Appendix F.

**Length (Number of words).** We calculated the average output word count for models that consistently completed all subtasks, achieving at least an 80% completion rate in sub-scenarios, excluding data from unsuccessful attempts. Most models substantially exceeded previous benchmarks for long-form generation tasks in terms of output length. Notably, the `LongWriter` (Bai et al., 2024) model excelled, efficiently meeting word count requirements for each subtask. Given the results and the weighted average score (wAvg) at a sequence length of 16K, the open-source `Qwen2-72B` and the closed-source `GPT-4o` models demonstrated optimal performance. At a sequence length of 32K, the `Llama3.1-8B` model, outperformed models with larger parameters, highlighting its efficiency in managing extended lengths.

## 3.3 ACCURACY TREND WITH VARYING SEQUENCE LENGTH

As illustrated in Figure 2, there is a clear decline in model performance as output length increases. Models exhibited strong adherence to initial instructions at shorter sequence lengths, but performance gradually degraded as the text generation extended beyond the 4,000-token threshold. This degradation aligns with trends identified in the NIAH dataset and underscores the challenge of maintaining instruction adherence and coherence over long outputs.

Figure 2: The right side of the figure illustrates the model's performance on specific instruction tasks at 16K as sequence length increases, whereas the left side depicts performance at 32K. All curves have been smoothed with a Moving Average.

This deviation becomes particularly pronounced when outputs exceed 4,000 tokens, where adherence to instructions significantly diminishes, and further deterioration is observed as outputs approach 16,000 tokens. In contrast, tasks involving shorter outputs, such as those in the NIAH dataset or simpler multi-needle tasks, showed near-perfect performance, highlighting the disparity in model behavior across different sequence lengths.

Potential reasons for this decline include limitations in the self-attention mechanism used in transformers, which may struggle to maintain meaningful context over long sequences. Additionally, models trained with limited long-form data may overfit to shorter patterns, leading to a loss of coherence in extended generations. These findings suggest that architectural changes or improved training strategies may be necessary to overcome these challenges in future iterations of LLMs.

## 3.4 THREE SPECIFIC TASK INSTRUCTIONS

Figure 3a presents the model's performance metrics across various task types: single, range, and periodic. The model demonstrated comparable proficiency in both single and range tasks, reflecting its capability to follow direct and straightforward instructions effectively. However, the slight reduction in performance for range tasks suggests that additional complexity, such as processing multiple data points within a defined range, introduces a marginal increase in cognitive load for the model.

The most significant decline in performance was observed in periodic tasks, where the model struggled to interpret instructions that required recurring events, such as "every four weeks starting from week 10." These tasks demand a higher degree of reasoning and temporal awareness, which may challenge the model's capacity to maintain consistency over extended sequences. As a result, outcomes for periodic tasks were considerably poorer compared to single and range tasks, which have clearer and more well-defined parameters. The model's performance hierarchy can generally be summarized as $single > range > periodic$, highlighting the increased difficulty associated with periodic tasks. This trend underscores the need for future improvements in long-context models, particularly in handling more complex, time-based instructions.

## 3.5 COMPARISON WITH LONG-CONTEXT INPUT

We examine the relationship between a model's ability to handle long inputs and its performance on long outputs. Specifically, we investigate whether a model's capacity to manage long-range inputs corresponds to improved performance on long-range outputs. For this analysis, we use the RULER dataset, a synthetic benchmark designed to flexibly configure sequence length and task complexity, making it ideal for comprehensive evaluations of long-context LLMs. We compare the models' performance on sequences of the same length, as shown in Figure 3b, which indicates a *significant performance gap* between input handling and output performance. At 16K tokens, the Pearson correlation coefficient is 0.51, while at 32K tokens, it increases to 0.66, suggesting that there is some overlap in the skills required for managing long inputs and generating long outputs, but these tasks are not entirely equivalent.

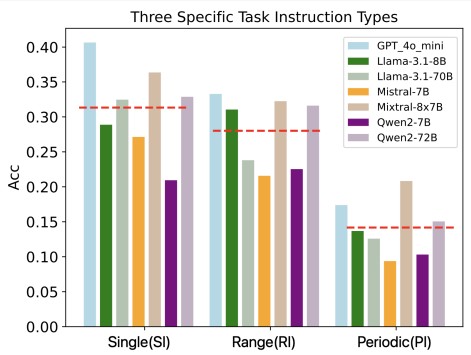
(a) Performance Comparison on three tasks settings

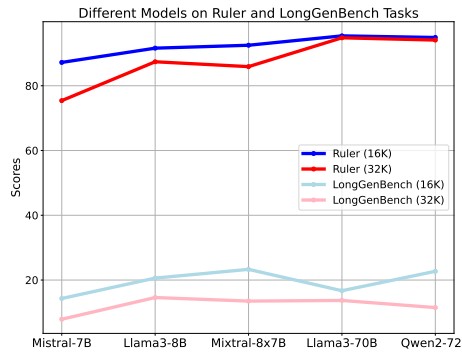
(b) Performance Comparison on Ruler and *LongGenBench* Tasks

Figure 3: The left Fig displays the models' performance on three different task settings, with the red line representing the average for each category. The right fig shows the performance and correlation of the Ruler and *LongGenBench* at the same length settings.

Handling long inputs primarily requires the model to retain and process existing information, while generating long outputs demands more complex reasoning, memory retention, and coherence management over extended sequences. Thus, models that excel in long-input retrieval may still struggle with long-form generation, particularly in tasks requiring strict instruction adherence over time. This distinction highlights the need for models to be optimized for both input handling and output generation to achieve consistent performance in long-context tasks.

## 4 ANALYSIS AND LIMITATIONS

**Richness of Content.** Despite efforts to design sub-scenarios that enhance task diversity and richness, the model's outputs tend to converge as output volume increases. This results in a homogenization of recorded events, even when differences in time and location should introduce variety. Such convergence not only degrades overall performance but also diminishes the diversity of the generated content, leading to repetitive and predictable outputs. In our experiments, approximately 45% of long outputs exhibited significant repetition, even when the model was given varied time or location prompts. Adjusting parameters like `repetition_penalty` during inference has shown limited success in mitigating this issue, highlighting the need for more advanced techniques to maintain content richness over long sequences.

**Rationality of Content.** While our current research focuses primarily on evaluating instruction-following capabilities, a more comprehensive analysis of content rationality and coherence is needed. For example, when tasked with generating a diary, the model should ensure that all recorded activities align with the specified careers. However, in many instances, this logical consistency is lacking. Additionally, temperature records in virtual diary entries often fail to reflect realistic temporal changes. For instance, in a San Francisco's diary task, we would expect temperatures to vary from cooler (0-10 degrees Celsius) at the beginning of the year to warmer (20-30 degrees Celsius) by mid-year. Yet, the model consistently generates warmer temperatures throughout, even into December. These issues may arise due to the model's limited exposure to temporally varied datasets, particularly in diary or climate-related contexts. Future work could address this by incorporating more domain-specific and temporally annotated data during fine-tuning.

**Instruction Data.** A significant performance discrepancy between models' abilities to handle long-range inputs (such as Ruler (Hsieh et al., 2024)) and their long-form output generation can likely be attributed to the length distribution of instruction-tuning data. Most instruction-tuning datasets are brief, typically under 200 tokens, and lack the extended instructional content necessary for generating longer outputs. This gap suggests that organizing or synthesizing instruction-tuning data with longer, more comprehensive examples could be a valuable direction for future research. Potential solutions

include applying transfer learning techniques from models trained on long-form datasets or using data augmentation methods to synthesize longer instructional content from existing short-form data.

**Generalizability.** LongGenBench effectively evaluates instruction-following in creative and technical tasks but may not fully capture the creativity and specialized knowledge required for abstract reasoning or unconstrained storytelling. Future versions could include open-ended tasks like creative fiction writing and legal document drafting, which demand intricate narratives and precision. Expanding in this direction would enhance the benchmark's versatility while providing deeper insights into LLMs' capabilities. However, LongGenBench's current focus on instruction adherence offers a strong foundation for evaluating practical, instruction-driven long-form text generation.

## 5 RELATED WORK

**Instruction Following.** Recent advances in instruction tuning models (Ouyang et al., 2022; Rafailov et al., 2024; OpenAI, 2022; Taori et al., 2023; Chiang et al., 2023) have underscored the need for scalable evaluation methods. LLMs have been used as evaluators, showing better alignment with human judgments than traditional metrics like BLEU (Papineni et al., 2002). However, LLM evaluations suffer from biases, such as sensitivity to presentation order and preference for verbose outputs (Wang et al., 2024; Pezeshkpour & Hruschka, 2023; Zheng et al., 2023). To mitigate these biases, meta-evaluation benchmarks like FairEval, MT-Bench, and LLMEval[2] (Wang et al., 2024; Zheng et al., 2023; Zhang et al., 2023) have been proposed. While recent studies have focused on improving LLM evaluations with diverse strategies (Zheng et al., 2023; Li et al., 2023; Zhang et al., 2023; Chan et al., 2023), they typically do not address longer context lengths.

**Long-context Benchmarks and Tasks.** Existing benchmarks focus on models handling long inputs. For instance, ZeroSCROLLS (Shaham et al., 2023) and LongBench (Bai et al., 2023) tackle tasks like long-document QA and query-based summarization. Synthetic benchmarks, like NeedleBench (Li et al., 2024) and Ruler (Hsieh et al., 2024), offer better control over variables such as sequence length and complexity. NeedleBench introduces the Ancestral Trace Challenge (ATC), while Ruler evaluates models across tasks like NIAH and multi-hop tracing. However, these benchmarks largely focus on input comprehension and do not assess long-form text generation, which is the primary focus of *LongGenBench*.

**Long-form Text Generation.** Research in long-form generation spans applications like story generation (Fan et al., 2019c; Xu et al., 2020), paragraph completion (Kang & Hovy, 2020), sustained conversation (Xu et al., 2022), and comprehensive QA (Fan et al., 2019a; Dasigi et al., 2021; Stelmakh et al., 2022; Lee et al., 2023). However, existing models and evaluation methods (Liu et al., 2023; Chiang & Lee, 2023; Liu et al., 2024; Bai et al., 2024) face challenges in maintaining quality over long outputs, often being limited by shorter text lengths (typically under 2000 tokens) (Shen et al., 2023). Recent work (Tan et al., 2024) seeks to improve evaluation criteria, but the gap between model capabilities and benchmark text lengths remains. In contrast, *LongGenBench* evaluates models on their ability to handle much longer sequences, with tasks requiring adherence to instructions over extended outputs (16K+ tokens).

## 6 CONCLUSION

We introduced *LongGenBench*, a synthetic benchmark that evaluates long-form generation capabilities of language models by testing their ability to follow instructions over extended sequences. In evaluating nine advanced models with context sizes ranging from 32K to 128K tokens, we observed significant performance degradation compared to benchmarks like "*Ruler*" with common failure modes including premature task termination, incomplete responses, disregard for instructions, and repetitive content generation. These results highlight key challenges for current models in handling long-form tasks and underscore the need for advancements in model architecture and training data to improve coherence, instruction adherence, and content diversity over extended outputs.

ACKNOWLEDGMENT

This research/project is supported by the Ministry of Education, Singapore, under its Academic Research Fund (AcRF) Tier 1 Grant, and funded through the SMU-SUTD Internal Research Grant Call.

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

## A  Symbol Definitions and Descriptions

This table 4 presents definitions and descriptions for various symbols used in task-related contexts, providing an overview of the key terminologies and their roles.

| Symbol | Definition | Description |
|---|---|---|
| $T$ | Main Task | The primary goal or task to be completed, such as designing a skyscraper or writing a diary. |
| $T_i$ | Subtask | A smaller portion of the main task, each responsible for a specific part, e.g., designing a specific floor. |
| $T_S$ | Single Instruction Task | A task requiring the model to inject specific information at a unique point in the generated text. |
| $T_R$ | Range Instruction Task | A task requiring the model to incorporate information within a specified range of the generated content. |
| $T_P$ | Periodic Instruction Task | A task that distributes specific information at predetermined intervals throughout the text. |
| $T_{S_i}$ | Single Instruction Task i | Represents an individual task from the Single Instruction Task set, focusing on a specific point in the text. |
| $T_{R_i}$ | Range Instruction Task i | Represents an individual task from the Range Instruction Task set, applied across a specific range. |
| $T_{P_i}$ | Periodic Instruction Task i | Represents an individual task from the Periodic Instruction Task set, recurring periodically throughout the text. |
| $CR$ | Completion Rate | The percentage of successfully completed subtasks out of the total number of subtasks, used to evaluate task performance. |
| $STIC-1$ | Specific Task Instruction Completion-1 | Evaluates how well the model follows specific task instructions, including Single, Range, and Periodic Instructions. Focuses on whether the instructions are executed correctly. |
| $STIC-2$ | Specific Task Instruction Completion-2 | Provides a more granular assessment, measuring not only adherence to instructions but also the consistency of execution throughout all subtasks. It looks at both presence and execution quality. |
| $A$ | Answer | Represents the complete response generated by the model for the main task. |
| $A_i$ | Subtask Answer | Represents the specific answer or output generated for an individual subtask, corresponding to $T_i$. |

Table 4: Symbol Definitions and Descriptions

## B  PROMPT TEMPLATES FOR FOUR TASK SCENARIOS

Below are the example templates for the four task scenarios: *Diary Writing*, *Menu Design*, *Skyscraper Design*, and *Urban Planning*. In the *Diary Writing* template, professions and names are customizable variables, allowing for flexibility in the generated content.

**Diary for 2018**

Emma is a photographer with a passion for chronicling her vibrant life through weekly diary entries. Captures:
1) Single Instruction (SI): Birthdays of family members, wedding anniversaries, etc.;
2) Range Instruction (RI): Family beach vacation in Maui, Week-long road trip across the Pacific Coast Highway, etc.;
3) Periodic Instruction (PI): Attend golf lessons at the local club, Join a weekend hiking group, etc.;
4) Weekly updates on weather changes, work developments, family life, and other interesting topics.
5) Use '#*#' to separate each weekly entry (e.g. example)
Generate a complete weekly diary for Emma for the entire year of 2018. Start from January 1st, a Monday, marking the first week, and continue through to December 31st, the end of the 52nd week. Ensure that the diary consists of 52 entries, one for each week. Each diary entry should be at least 200 words. When the design of all 52 weeks is complete, use '*** finished ***' to indicate the end of the document. Ensure clarity and continuity without any interruptions or omissions in the narrative throughout the year.
*** started ***
#*# Week 1 (January 1st - January 7th):

**Menu for 2018**

As Chef Roy, a world-renowned chef at a globally renowned restaurant, you are tasked with designing an entire year's menu for 2018. Your menu should be varied and innovative, adhering to the following guidelines:
1) Single Instruction (SI): ("Independence Day Celebration", "2018-07-04", "American Apple Pie"), ("Summer Solstice Celebration", None, "Midsummer Night's Fish Fry"), etc;
2) Range Instruction (RI): ("Mushroom Season Specials", "Various Mushroom Dishes"), ("Seafood Season Extravaganza", "Fresh Seafood Platter"), etc ;
3) Periodic Instruction (PI): ("Seafood Fridays", 2, "Fish and Chips"), ("Monthly Steak Night", 3, "Prime Ribeye Steak"), etc.;
4) Use '#*#' to separate each weekly menu (e.g. example) Generate a comprehensive weekly menu diary for the entire year of 2018, start from January 1st, a Monday, marking the first week, and continuing until December 31, the end of the 52nd week. Ensure that the diary consists of 52 entries, one for each week. Each weekly menu must include a detailed description of the offerings, featuring at least two options for appetizers, main courses, side dishes, desserts, and drinks. Ideally, between 200 and 220 words per menu description to ensure thoroughness and richness of detail. Conclude the diary with '*** finished' to signify the completion of the year's menu planning. Ensure clarity and continuity without any interruptions or omissions in the menu throughout the year.
*** started ***
#*# Menu Week 1 (January 1st - January 7th):",

**Skyscraper Design**

Construct a skyscraper with 100 floors. Please follow the detailed floor assignments below:
1) Single Instruction (SI): office, conference room, retail store, etc;
2) Range Instruction (RI): hospital with various departments, corporate headquarters for a major company, etc ;
3) Periodic Instruction (PI): outdoor terrace, sky garden, etc.;
4) Document each floor independently with detailed descriptions of the intended facilities, architectural features, and unique design elements.
5) Use '#*#' to separate the documentation for each floor (e.g. example).
Ensure that the document consists of 100 entries, each containing at least 150 words. Ensure clarity and continuity without any interruptions or omissions in the narrative throughout the document. When the design of all 100 floors is complete, use '*** finished' to indicate the end of the document.
*** started ***
#*# Floor 1:

**Urban Planning**

Design a vibrant and diverse city using a 10x10 block grid, numbered from 1 to 100. Arrange the blocks sequentially from left to right and top to bottom. Ensure that each block is uniquely planned to reflect a wide array of city facilities, highlighting the rich urban environment and cultural diversity.
1) Single Instruction (SI): theater, museum, etc.;
2) Range Instruction (RI): shopping district, industrial park, etc.;
3) Periodic Instruction (PI): public restroom, convenience store, etc.;
4) Document each block independently with detailed descriptions of the intended facilities, architectural features, and unique design elements.
5) Use '#*#' to separate the documentation for each block like (e.g.example)
Ensure that the document consists of 100 entries, each containing at least 150 words. Ensure that the document contains detailed descriptions for each block, with a minimum of 150 words per description. Ensure clarity and continuity in the narrative throughout the document without any interruptions or omissions. When all block assignments are complete, use '*** finished' to indicate the end of the document.
*** started ***
#*#Block 1 (0, 0):

## C    EVALUATION PIPELINE

The evaluation pipeline is designed to systematically assess the ability of long-context language models (LLMs) to follow specific, complex instructions. The process can be summarized in three key steps:

### STEP 1. GENERATION OF OUTPUTS FROM THE LONG-CONTEXT LLM

Given an input task ($T$) that describes a set of instructions, we prompt the LLM to generate detailed outputs. The output ($A$) comprises a list of descriptions, represented as:

$$A = \{A_1, A_2, \ldots, A_n\}$$

**Example: Given the prompt (ref Appendix B)**

> **Construct a skyscraper with 100 floors.** The floor assignments are detailed as follows:
>
> - **Specific floor requirement:** Designate Floor 11 for a small art gallery.
> - **Range floor requirement:** Allocate Floors 32 to 39 for corporate headquarters of a major company.
> - ...

The LLM generates a response describing each floor in detail, such as:

**Answer:**

> - Floor 1: ... Lobby ...
> - Floor 11: ... Small art gallery ...
> - Floor 32: ... Corporate headquarters ...
> - Floor $n$: ...

### STEP 2. EXTRACTING AND MATCHING RELEVANT FLOOR ASSIGNMENTS (CHECK SET)

From the initial input ("T"), we create a **check set** containing specific floor assignments to verify if the LLM correctly follows the instructions.

For the example above, the check set includes:

> **Check Set:**
>   - Floor 11: Small art gallery
>   - Floor 32: Corporate headquarters
>   - Floor 33: Corporate headquarters
>   - . . .

We then extract the relevant parts of the LLM output ("A") that correspond to the floor assignments described in the check set.

STEP 3. EVALUATION USING LLAMA 3.3-70B INSTRUCTION MODEL

For each extracted pair, we use the Llama 3.3-70B model to evaluate whether the output ("$A_i$") for a given task segment ("$T_{si}$") has correctly fulfilled the specified instruction.

This evaluation task is framed as a simple **binary classification** problem, which aims to determine if the specific instruction was fulfilled ("yes" or "no"). The prompt used for this evaluation is as follows:

> **Evaluation Prompts**
>   - *Example 1*: XXXX **Answer:** Analysis + #*# Yes
>   - *Example 2*: XXXX **Answer:** Analysis + #*# No
>
> **Context:** Long-context model output: *"Floor 11: . . . small art gallery . . . "*
> **Instructions:** Does this context include *'small art gallery'*?
> **Answer:** Please refer to the above example, provide your analysis, and respond with either #*# Yes or #*# No.

Notably, this binary evaluation is straightforward. We manually labeled 300 data points, and the model's output matched human evaluations for all cases.

By using this process, we transform the evaluation of long-context text generation into multiple evaluations of smaller segments. This enables systematic and thorough verification of how well the LLM follows the instructions for each specific task (as detailed in the check set).

# D    MODELS

In this benchmark, we evaluated ten LLMs, including both open-source and closed-source models. These models vary in parameter size and context length capabilities, which are crucial factors in their performance on long-form text generation tasks. The key details for each model are outlined in Table 5. These include closed-source models like GPT-4o-mini and GPT-4o, which support a context length of 128K tokens and serve as state-of-the-art baselines for long-context handling. Open-source models, such as Llama3.1-8B and Llama3.1-70B, offer similar context lengths and represent the latest in large-scale, open-access LLMs. Qwen2-7B and Qwen2-72B, developed by Qwen, also support 128K tokens and handle complex long-text tasks. Additionally, we evaluated Mixture of Experts (MoE) models like Mistral-v0.2 and Mixtral-8x7B, both with context lengths of 32K tokens, focusing on memory efficiency and scalability. FILM-7B, designed for creative and technical tasks, supports 128K tokens and excels in generating detailed, context-rich content. Finally, Longwrite-llama3.1-8B, based on Llama3.1, is optimized for long-form narrative tasks with a context window of 128K tokens. Together, these models offer a diverse representation of advancements in long-context LLMs, showcasing their ability to handle long-form, instruction-driven generation tasks.

Table 5: Information of evaluated and analyzed models in *LongGenBench*.

| Model | Aligned | Size | Context Length | Huggingface (Wolf et al., 2019) / API |
|---|---|---|---|---|
| GPT-4o-mini (OpenAI, 2024a) | ✓ | - | 128K | `gpt-4-mini` |
| GPT-4o (OpenAI, 2024b) | ✓ | - | 128K | `gpt-4o-2024-08-06` |
| Llama3.1-8B-Instruct (Dubey et al., 2024) | ✓ | 8B | 128K | meta-llama/Meta-Llama-3.1-8B-Instruct |
| Llama3.1-72B-Instruct (Dubey et al., 2024) | ✓ | 70B | 128K | meta-llama/Meta-Llama-3.1-70B-Instruct |
| Qwen2-7B-Instruct (Yang et al., 2024) | ✓ | 7B | 128K | Qwen/Qwen2-7B-Instruct |
| Qwen2-72B-Instruct (Yang et al., 2024) | ✓ | 72B | 128K | Qwen/Qwen2-72B-Instruct |
| Mistral-v0.2 (Mistral.AI, 2023) | ✓ | 7B | 32K | mistralai/Mistral-7B-Instruct-v0.2 |
| Mixtral-8x7B (Jiang et al., 2024) | ✓ | 8x7B | 32K | mistralai/Mixtral-8x22B-Instruct-v0.1 |
| FILM-7B (An et al., 2024) | ✓ | 7B | 128K | In2Training/FILM-7B |
| Longwrite-llama3.1-8B (Bai et al., 2024) | ✓ | 8B | 128K | THUDM/LongWriter-llama3.1-8b |

# E    EXPLANATION OF METRICS

## E.1    STIC-1 AND STIC-2

We appreciate the feedback and have provided an example using the results from Table 3 of our experiments (specifically comparing LLaMA3.1-8B and Qwen2 under the short-version setting).

As shown in Table 6, Qwen2's STIC-1 score is higher than that of LLaMA3.1-8B, while its STIC-2 score is lower. This difference can be attributed to the Completion Rate (CR) of each model. Qwen2 has a significantly lower CR compared to LLaMA3.1-8B. Specifically, Qwen2 typically achieves around 60% completion for tasks (e.g., designing a 100-story skyscraper but stopping at roughly 60 stories). On the other hand, LLaMA3.1-8B generally completes around 93 layers (93% completion).

In the case of STIC-1, we are evaluating the correctness of the output based on the number of layers that are actually generated. Qwen2 demonstrates a higher completion rate when the denominator consists of the 60 layers it has output (compared to LLaMA3.1-8B, which has a denominator of 93 layers).

For STIC-2, however, we consider the entirety of the expected output. Since Qwen2 lacks the remaining 30 layers, the STIC-2 score is lower when the denominator becomes the entire requirement (as the missing output significantly affects its score).

As mentioned in our paper, STIC-2 is designed to take into account a more comprehensive perspective on output completeness. We are considering simplifying our metrics by using only STIC-2, as it may be easier to understand and provide a more holistic evaluation.

Table 6: Comparison of STIC-1 and STIC-2 Scores

| Model | Length | CR | STIC-1 | STIC-2 |
|---|---|---|---|---|
| LLaMA3.1-8B | 128K | 93.5% | 23.4% | 22.0% |
| Qwen2-7B | 128K | 60.0% | 27.9% | 16.1% |

## E.2    EXAMPLE EXPLANATION

To further illustrate the concepts, we have constructed a 3-level building for illustration[8]:

- Consider a **3-level building** with the following constraints:
    - $T_{S_1}$: *"Floor 1 must have a coffee shop."*
    - $T_{S\_2}$: *"Floor 1 must have a reception desk."*
    - $T_P$: $\{T_{P_1}, T_{P_2}, T_{P_3}\}$, where each $T_{P_i}$ means *"Floor $i$ must have a washroom."*

The model generates the following output:

---

[8]This example is adapted from ICLR reviewer bHUs. We are deeply grateful for their insightful comment and in-depth discussion, which significantly improved the clarity of this paper.

"floor1: coffee shop, washroom; floor2: washroom."

In this scenario, the **check_set** is $\{T_{S_1}, T_{S_2}, T_{P_1}, T_{P_2}, T_{P_3}\}$. Note that $T_P$ applies to all three floors, requiring separate evaluation for each $TP_i$.

With the current model output, the **completion rate (CR)** for the main task is $2/3$. Although the task requires outputs for three floors, the model only provided outputs for two floors.

For **STIC-1**, we consider how accurately the model has outputted information at the floor level. Since the model output only contains two floors, we evaluate the constraints for these two floors to determine if they are fully met. For these two floors, the constraints are $T_{S_1}, T_{S_2}, T_{P_1}, T_{P_2}$, totaling 4 constraints. The model has correctly fulfilled 3 out of these 4 requirements, resulting in **STIC-1** of $3/4$.

For **STIC-2**, we evaluate the entire **check_set**, which consists of $T_{S_1}, T_{S_2}, T_{P_1}, T_{P_2}, T_{P_3}$. The model has fulfilled 3 out of these 5 requirements, so **STIC-2** equals $3/5$.

The distinction between **STIC-1** and **STIC-2** allows identification of the specific reasons for any drop in performance. It helps determine whether the issue lies in the model's inability to follow instructions for a given output or whether it lacks a complete output in the first place. For example, in the case of a lower **STIC-2**, the low score may be due to incorrect outputs for some floors or due to incomplete outputs for the floors. In such cases, **CR** and **STIC-1** can be used together to further evaluate and make informed judgments.

## F  ERROR ANALYSIS

We showcase two error generated output that highlight the common issues faced by large language models in maintaining coherence and instruction-following over extended sequences. In the Diary for 2018 scenario, while the model initially adheres to the instructions for weeks 6 and 7, it progressively diverges from the given timeline, misplacing events and generating off-topic content by week 15 and beyond. Similarly, in the Skyscraper Design task, the model starts with correct descriptions for specific floors, such as Floor 11 designated as an art gallery, but it fails to maintain consistency with the subsequent instructions, particularly in the range and periodic requirements for floors 32-39 and every 15th floor. These examples illustrate the challenges long-context models face in sustaining adherence to task instructions, particularly as sequence length increases.

**Diary for 2018**

Noah is a photographer with a passion for chronicling his vibrant life through weekly diary entries. Captures in daily:
1) Family member birthday: husband (yourself) (birthday on March 08), wife (birthday on July 09), child_1 **(birthday on November 06)**.
2) Joining a fitness and wellness retreat in Bali in week **6-7**.
3) Join a local running club for weekend marathons every **5** weeks on weekends, starting from week **10**.
.....
============================= Output =============================
.....
Week 6 (February 5th - February 11th): Noah had joined a fitness and wellness retreat in Bali in this week. .... ✓
Week 7 (February 12th - February 18th): Continuing in Bali, I spent the week participating in various workshops and seminars. .... ✓
.....
Week 15 (April 9th - April 15th): The family is in full spring mode, with kids spending time outside, exploring, and enjoying the nice weather. ..... ✗
.....
Week 35 (August 27th - September 2nd): The summer heat has subsided, and we've been enjoying the milder weather 2013 going on hikes, playing in the park, and exploring our neighborhood. ....
✗
.....
Week 45 (November 5th - November 11th): The kids have been enjoying their school routines, and we've been making the most of our evenings together. ..... ✗

**Skyscraper Design**

Construct a skyscraper with 100 floors. Please follow the detailed floor assignments below:
1) Specific floors requirements: Designate Floor **11** for small art gallery use. ....
2) Range floors requirements: Allocate Floors **32 to 39** for corporate headquarters for a major company.
3) Periodic floors requirements: Include a sky garden every **15** floors, starting from Floor **30**.
.....
============================= Output =============================
.....
Floor 11: Designated for art gallery use, Floor 11 is a sophisticated and flexible space designed to celebrate visual arts..... ✓
.....
Floor 32: Floor 32 serves dual purposes, housing a renowned photography studio and corporate offices. ..... ✗
.....
Floor 34: Transitioning into a leisure space, Floor 34 hosts a small cinema, providing an exclusive entertainment venue within the skyscraper. ..... ✗
.....
Floor 60: This floor houses a luxury watch and timepiece atelier, celebrating the art of horology and fine craftsmanship. ..... ✗
.....
Floor 90: Floor 90 offers a dynamic e-commerce and digital marketing center focused on online business innovation and consumer engagement strategies. ..... ✗

## G  DIFFERENT PROMPT FORMAT COMPARE

The two prompt formats differ primarily in the structure and arrangement of the instructions within the prompt. In Prompt - 1, the order follows a sequence of Single Instruction (SI), Range Instruction (RI), and Periodic Instruction (PI). Conversely, in Prompt - 2, this order is altered by swapping the positions of SI, RI, and PI. Additionally, the Generate prompt, which is a critical component of the task, was rewritten in Prompt - 2 by GPT-4.

From the table, it is evident that different prompt formats influence the performance metrics of the models, such as CR, STIC-2, length, and wAvg. For instance, in Prompt - 1, the Mistral-7B-Instruct model achieves the highest CR (81.8) and STIC-2 (17.44%), while LongWriter-llama3.1-8b lags behind with a CR of 46.0 and STIC-2 of 9.83%. Similarly, under Prompt - 2, the same trend is observed: Mistral-7B-Instruct maintains its lead with a CR of 62.3 and STIC-2 of 16.29%, while LongWriter-llama3.1-8b again ranks lowest with a CR of 24.3 and STIC-2 of 8.35%.

Although the prompt format does affect the absolute values of these metrics (e.g., all models show reduced CR under Prompt - 2 compared to Prompt - 1), the relative rankings remain unchanged. This consistency suggests that while prompt design impacts performance, it does not alter the comparative effectiveness of the models.

| Prompt Format | Model | CR | STIC-2 | Length (word) | wAvg | Rank |
|---|---|---|---|---|---|---|
| Prompt - 1 | LongWriter-llama3.1-8b | 46.0 | 9.83% | 11036 | 4.5 | 3 |
| | Qwen2-7B-Instruct | 60.0 | 16.13% | 5138 | 9.7 | 2 |
| | Mistral-7B-Instruct-v0.2 | 81.8 | 17.44% | 7296 | 14.3 | 1 |
| Prompt - 2 | LongWriter-llama3.1-8b | 24.3 | 8.35% | 6189 | 2.0 | 3 |
| | Qwen2-7B-Instruct | 57.3 | 16.34% | 4334 | 9.4 | 2 |
| | Mistral-7B-Instruct-v0.2 | 62.3 | 16.29% | 4750 | 10.2 | 1 |

Table 7: Model comparison with different prompt formats

# H    EXPLANATION OF UNIT DIFFERENCES: TOKENS VS. WORDS

In this work, the term *"16K/32K"* refers to the required number of tokens for the model output, adhering to the standard conventions when discussing model context lengths. However, for evaluating the actual generated output, we employed word count as the measurement unit. This distinction was made for the following key reasons:

**Variability in Token-to-Word Conversion**: Different tokenizers vary in how they convert tokens into words, typically resulting in an average ratio of about 1.5 tokens per word. Therefore, the actual word count of the output is usually approximately two-thirds of the target token length. This variability makes word count a more consistent measure for analyzing content.

**Emphasis on Content Quality**: Our primary focus was on evaluating the quality and completeness of the generated content. Word count provides a more straightforward perspective on the substance of the output, which is crucial for content assessment.

