# OpenReview forum: "LongGenBench: Benchmarking Long-Form Generation in Long Context LLMs"
_ICLR.cc/2025/Conference — ICLR 2025 Poster_

### Official Review · Reviewer_bHUs · 2024-10-24

**Soundness:** 3
**Presentation:** 2
**Contribution:** 4
**Rating:** 8
**Confidence:** 5

**Summary:**

The paper proposes a new benchmark for long-form generation where the model-generated content, rather than the input context, is long. Specifically, the model is asked to roll out a detailed description of certain tasks such as diary over a year or floor plan for a skyscraper, subject to a few specific instructions that can be either singular or recurrent. Evaluation metrics include main task completion to test whether generation follows the expected format, as well as the success rate for specific instructions at both micro and macro level. Results demonstrate that the proposed benchmark is much more challenging than needle-in-the-haystack tasks for long context evaluation. Models generally do not perform very well within their claimed context lengths.

**Strengths:**

* The paper is the first to study long-form generation as opposed to long-context generation. The perspective is novel, interesting, and of practical value to unleash the potential of LLMs for more complicated tasks.
* Problems in the benchmark are constructed in a sound and intuitive way. While evaluating the quality of long text snippets is usually challenging and complex, the smart design in this paper enables reliable and accurate evaluation for long-form capability.

**Weaknesses:**

One of my major concerns is the clarity of writing in the evaluation part.
* The definition of STIC-1/STIC-2 isn't quite clear. Using the notation $T_S=(T_{S_1}, T_{S_2}, \dots)$ in Sec 2.3, my best guess is that STIC-1 means the average success rate over $(T_{S_1}, T_{S_2}, \dots)$, while STIC-2 counts the entire $T_S$ as successful only if all $(T_{S_1}, T_{S_2}, \dots)$ are successful, and gives 0 score otherwise.
* The abbreviation **CR** in Table3 isn't defined anywhere, though I can guess this is likely the Completion Rate in Main Task Completion.
* The terms "main task", "subtask",  "instruction task", "specific task" are used in a confusing way. It would be very helpful to unify them with clear definitions, and to associate the terms to symbols like $T_S$ or $T_{S_1}$.
* Missing x-ticks in Figure 2.

Apart that, there are a few technical details that are unclear to me.
* How do you determine whether the generated text at one specific point satisfies all task requirements? For example, given the generated diary of a particular day, how do you verify that all required activities (e.g. wedding, vacation, and etc.) are covered? I would imagine that a natural language inference (NLI) module should be involved but it's not mentioned in the paper.
* In Table 3, though the length to be evaluated at is 16K/32K respectively, the actual generation length seems only around half of the max length. How are the 16K/32K defined?

**Questions:**

* In L460-468, it is mentioned that there are significant repetitions in long-form generations. Are these repetitions correct with respect to the given instructions, or are they semantically wrong (i.e. violating the given instructions)? The former indicates that it's probably caused by how instructions in the prompt are designed, while the latter means that model struggles at producing long and meaningful generation.

---

> ### Author Response · Authors · 2024-11-18
> **Comment (1/4)**
>
> Thank you for your constructive review and valuable suggestions! Below, we provide a detailed response to your questions and comments. If any of our responses fail to sufficiently address your concerns, please inform us, and we will promptly follow up.
>
> **W1: Differences between STIC-1 and STIC-2**
>
> Thank you for your insightful comments regarding the need for an example to illustrate the differences between STIC-1 and STIC-2. We appreciate your feedback and have included a comparative example in the revised manuscript, specifically referencing results from Table 3 of our experiments, which compare LLaMA3.1-8B and Qwen2 under the short-version setting.
> STIC-1 and STIC-2 are designed to evaluate instruction adherence at different levels of granularity:
>
> - **STIC-1** measures the average success rate of individual instructions within the actual completed portion of a task. For instance, STIC-1 evaluates the correctness of each generated output segment relative to the portion of the task completed, without penalizing for incomplete task segments. This allows STIC-1 to reflect the model’s consistency in following instructions within its generated content.
> - **STIC-2**, in contrast, provides a comprehensive assessment of output completeness. This metric evaluates a task as a whole, counting it as successful only if all specified instructions across the full task length are followed correctly. STIC-2 thus captures the model’s ability to handle long and complex tasks comprehensively, without any partial completion.
>
> **Example Comparison**
>
> The table below compares LLaMA3.1-8B and Qwen2 to illustrate how these metrics diverge:
>
> | Model         | Length | CR    | STIC-1 | STIC-2 |
> |--------|--------|-------|--------|--------|
> | LLaMA3.1-8B   | 128K   | 93.5% | 23.4%  | 22.0%  |
> | Qwen2-7B      | 128K   | 60.0% | 27.9%  | 16.1%  |
>
> In this case, Qwen2 achieves a higher STIC-1 score than LLaMA3.1-8B but a lower STIC-2 score. This difference arises from the models’ varying Completion Rates (CR). Qwen2 typically achieves a 60% completion rate, akin to completing approximately 60 floors of a 100-story skyscraper design task, while LLaMA3.1-8B completes closer to 93 floors.
>
> For **STIC-1**, Qwen2 scores higher since its evaluation is based only on the 60 floors it successfully generates, compared to LLaMA3.1-8B’s 93 floors. STIC-1 does not penalize Qwen2 for the missing floors, focusing instead on the instruction adherence within the portion generated. In contrast, **STIC-2** evaluates the completeness of the entire task; since Qwen2 does not generate the remaining 40 floors, its STIC-2 score is negatively impacted due to this incomplete output.
>
> We trust that this explanation clarifies the distinctions between STIC-1 and STIC-2, and we thank you for the opportunity to expand on these metrics in our revised manuscript.
>
>
> **W2: Abbreviation "CR" in Table 3**
>
> Thank you for pointing out that the abbreviation "CR" in Table 3 was not defined clearly. Your understanding is correct: "CR" stands for Completion Rate in Main Task Completion. This metric measures the proportion of the main task completed by the model within the given constraints, providing a quantitative indicator of progress and adherence to the task's full requirements.
>
> We have updated the corresponding section to clearly define "CR" upon its first occurrence to ensure clarity for all readers. We appreciate your attention to detail and your assistance in improving the overall quality of our work.
>
> **W4: Missing X-ticks in Figure 2**
>
> Thank you for pointing out the missing x-ticks in Figure 2. The x-axis represents the token count of the output length, and we will update Figure 2 to include x-ticks for clarity. We appreciate your attention to detail and your assistance in improving the overall quality of our work.

---

> > ### Author Response · Authors · 2024-11-18
> > **Comment (2/4)**
> >
> > **W3: Terminology Clarity**
> >
> > Thank you for your feedback regarding the use of terms like "main task," "subtask," "instruction task," and "specific task." We understand that these terms may create confusion when not clearly defined and consistently applied throughout the paper. To bring clarity, we have prepared a table that unifies these terms, defines each one, associates them with corresponding symbols, and provides an overall description. This should help create a unified understanding of the evaluation framework:
> >
> > | Symbol      | Definition                        | Description                                                                                           |
> > |-------------|-----------------------------------|-------------------------------------------------------------------------------------------------------|
> > | **T**       | Main Task                         | The primary goal or task to be completed, such as designing a skyscraper or writing a diary.          |
> > | **Tᵢ**      | Subtask                           | A smaller portion of the main task, each responsible for a specific part, e.g., designing a specific floor. |
> > | **TS**      | Single Instruction Task           | A task requiring the model to inject specific information at a unique point in the generated text.    |
> > | **TR**      | Range Instruction Task            | A task requiring the model to incorporate information within a specified range of the generated content. |
> > | **TP**      | Periodic Instruction Task         | A task that distributes specific information at predetermined intervals throughout the text.          |
> > | **Tsᵢ**     | Single Instruction Task i         | Represents an individual task from the Single Instruction Task set, focusing on a specific point in the text. |
> > | **TRᵢ**     | Range Instruction Task i          | Represents an individual task from the Range Instruction Task set, applied across a specific range.   |
> > | **TPᵢ**     | Periodic Instruction Task i       | Represents an individual task from the Periodic Instruction Task set, recurring periodically throughout the text. |
> > | **CR**      | Completion Rate                   | The percentage of successfully completed subtasks out of the total number of subtasks, used to evaluate task performance. |
> > | **STIC-1**  | Specific Task Instruction Completion - 1 | Evaluates how well the model follows specific task instructions, including Single, Range, and Periodic Instructions. Focuses on whether the instructions are executed correctly. |
> > | **STIC-2**  | Specific Task Instruction Completion - 2 | Provides a more granular assessment, measuring not only adherence to instructions but also the consistency of execution throughout all subtasks. It looks at both presence and execution quality. |
> > | **A**       | Answer                            | Represents the complete response generated by the model for the main task.                             |
> > | **Aᵢ**      | Subtask Answer                    | Represents the specific answer or output generated for an individual subtask, corresponding to Tᵢ.    |
> >
> > This table will be incorporated into the manuscript to define these terms upon first occurrence and to ensure consistent use throughout. We appreciate your attention to this detail, as it helps us improve the clarity and readability of our work.
> >
> > Thank you again for your suggestion, which has significantly contributed to improving the precision of our terminology.

---

> ### Author Response · Authors · 2024-11-18
> **Comment (3/4)**
>
> **W5:Evaluation of Correctness**
>
> Thank you for your questions and for prompting us to clarify our evaluation process further. We realize our original explanation may have lacked detail, so we are pleased to provide a more comprehensive breakdown of the evaluation pipeline here, as also outlined in Section 2.5 of the paper.
>
> Our evaluation pipeline systematically assesses the ability of long-context LLMs to follow specific, complex instructions. The process can be summarized in three key steps:
>
> ### 1. Generation of Outputs from the Long-context LLM
>
> Given an input task (`T`) that describes a set of instructions, we prompt the LLM to generate detailed outputs. The output (`A`) comprises a list of descriptions, represented as: `A = {A1, A2, ..., An}`
>
>
> **Example: Given the prompt (ref Appendix SCENARIO)**
> > **Construct a skyscraper with 100 floors.** The floor assignments are detailed as follows:
> > - **Specific floor requirement:** Designate Floor 11 for a small art gallery.
> > - **Range floor requirement:** Allocate Floors 32 to 39 for corporate headquarters of a major company.
> > - ...
>
> The LLM generates a response describing each floor in detail, such as:
> > - Floor 1: ... Lobby ...
> > -   ...
> > - Floor 11: ... Small art gallery ...
> > -   ...
> > - Floor 32: ... Corporate headquarters ...
> > -   ...
> > - Floor n: ...
>
> ### 2. Extracting and Matching Relevant Floor Assignments (Check Set)
>
> From the initial input ("T"), we create a **check set** containing specific floor assignments to verify if the LLM correctly follows the instructions.
>
> For the example above, the check set includes:
> > - Floor 11: Small art gallery
> > - Floor 32: Corporate headquarters
> > - Floor 33: Corporate headquarters
> > - ...
>
> We then extract the relevant parts of the LLM output ("A") that correspond to the floor assignments described in the check set.
>
> ### 3. Evaluation Using Llama 3.1-8B instruction Model
>
> For each extracted pair, we use the Llama 3.1-8B model to evaluate whether the output ("Ai") for a given task segment ("Tsi") has correctly fulfilled the specified instruction.
>
> This evaluation task is framed as a simple **binary classification** problem, which aims to determine if the specific instruction was fulfilled ("yes" or "no"). The prompt used for this evaluation is as follows:
>
> **Evaluation Prompts**
> > - *Example 1*: XXXX **Answer:** Analysis + #*# Yes
> > - *Example 2*: XXXX **Answer:** Analysis + #*# No
> > - **Context:** Long-context model output: *"Floor 11: ... small art gallery ..."*
> > - **Instructions:** Does this context include 'small art gallery'?
> > - **Answer:** Please refer to the above example, provide your analysis, and respond with either #*# Yes or #*# No.
>
> Notably, this binary evaluation is straightforward. We manually labeled 300 data points, and the model's output matched human evaluations for all cases.
>
> By segmenting the long-generation task into smaller units and evaluating each one individually, our approach offers a thorough and systematic method to verify instruction adherence across the full sequence. This ensures that the LLM’s performance on each component of the task can be accurately and efficiently assessed.
> We hope this detailed explanation clarifies our approach, and we thank you for the opportunity to elaborate on our evaluation methodology.
>
>
> **W6: 16K/32K Length Definition in Table 3**
>
> Thank you for pointing out the potential confusion around the “16K/32K” length designation in Table 3. To clarify, the term "16K/32K" refers to the required number of tokens for the model output, aligning with standard conventions when discussing model context lengths. However, for evaluating the actual generated output, we used word count as the measurement unit.
>
> This approach was taken for two key reasons:
>
> - Token-to-Word Conversion Variability: Different tokenizers can vary in how they map tokens to words, typically resulting in an average conversion ratio of approximately 1.5 tokens per word. Consequently, the actual word count of the output is generally around two-thirds of the target token length.
>
> - Practical Focus on Output Content: Since our evaluation prioritizes assessing the quality and completeness of the final output, using word count provides a clearer perspective on the generated content itself.
> We will revise the paper to make this distinction more explicit, ensuring clarity for readers regarding our choice of measurement and terminology. Thank you again for highlighting this area, as it allows us to improve the precision of our explanations.

---

> > ### Author Response · Authors · 2024-11-18
> > **Comment (4/4)**
> >
> > **Q1: Long-form Generations (L460-468)**
> >
> > Thank you for your question regarding the nature of repetitions in long-form generations. We appreciate your attention to this aspect of our evaluation.
> >
> > To clarify, the observed repetitions are indeed semantically incorrect and violate the specific instructions provided in the prompt. Our instructions required unique content for each floor, with explicitly defined elements for certain floors, such as a “sky garden” for Floor 60. However, the model-generated outputs exhibited significant repetition across multiple floors, disregarding these unique content requirements.
> >
> > For example, the outputs for Floors 59 to 62 are identical in both structure and content, repeating the following features verbatim:
> >
> > - "This floor features a contemporary design with a focus on functionality and comfort, creating a modern and efficient workspace. Here are the detailed features: 1) Reception and Waiting Area: ... 2) Administrative Offices: ... 3) Conference Rooms: ... 4) Pantry and Break Room: ... 5) Storage and Maintenance: ..."
> >
> > This repetition conflicts with our specific requirement for unique descriptions, particularly on Floor 60, where a "sky garden" was instructed but not included. Such repeated content indicates that the model struggles to produce diverse and meaningful long-form outputs that align with detailed instructions over extended text generation.
> >
> > These repetitions are not due to ambiguities or redundancy in the prompt design but instead reflect the model's challenges in adhering to complex, unique requirements over lengthy outputs. This behavior suggests a need for improved handling of detailed, long-context instructions within the model.
> >
> > Thank you again for raising this point, as it underscores an important limitation in the current state of long-context generation.
> >
> >
> > ***Example (LLaMa 3.1-70B output)***
> >
> > - Floor 59: This floor features a contemporary design with a focus on functionality and comfort, creating a modern and efficient workspace. Here are the detailed features: 1) Reception and Waiting Area: The reception area is designed with a sleek aesthetic, featuring a minimalist reception desk, comfortable seating, and stylish decorative elements. The area is well-lit and welcoming. 2) Administrative Offices: The office spaces are spacious and well-organized, with ample desk space, ergonomic chairs, and modern IT equipment. A central break area offers refreshments and a relaxed environment. 3) Conference Rooms: The floor features two well-equipped conference rooms, each with AV equipment and comfortable seating, ideal for meetings or client presentations. 4) Pantry and Break Room: A pantry area provides a selection of snacks and beverages, while a separate break room offers a more casual space for relaxation or quick breaks. 5) Storage and Maintenance: The floor features ample storage areas and maintenance facilities to support daily operations. \n",
> >
> > - Floor 60: This floor features a contemporary design with a focus on functionality and comfort, creating a modern and efficient workspace. Here are the detailed features: 1) Reception and Waiting Area: The reception area is designed with a sleek aesthetic, featuring a minimalist reception desk, comfortable seating, and stylish decorative elements. The area is well-lit and welcoming. 2) Administrative Offices: The office spaces are spacious and well-organized, with ample desk space, ergonomic chairs, and modern IT equipment. A central break area offers refreshments and a relaxed environment. 3) Conference Rooms: The floor features two well-equipped conference rooms, each with AV equipment and comfortable seating, ideal for meetings or client presentations. 4) Pantry and Break Room: A pantry area provides a selection of snacks and beverages, while a separate break room offers a more casual space for relaxation or quick breaks. 5) Storage and Maintenance: The floor features ample storage areas and maintenance facilities to support daily operations. \n",
> >
> > - Floor 61: This floor features a contemporary design with a focus on functionality and comfort, creating a modern and efficient workspace. Here are the detailed features: 1) Reception and Waiting Area: The reception area is designed with a sleek aesthetic, featuring a minimalist reception desk, comfortable seating, and stylish decorative elements. The area is well-lit and welcoming. 2) Administrative Offices: The office spaces are spacious and well-organized, with ample desk space, ergonomic chairs, and modern IT equipment. A central break area offers refreshments and a relaxed environment. 3) Conference Rooms: The floor features two well-equipped conference rooms, each with AV equipment and comfortable seating, ideal for meetings or client presentations. 4) Pantry and Break Room: A pantry area provides a selection of snacks and beverages, while a separate break room offers a more casual space for relaxation or quick breaks. ",

---

> > > ### Comment · Reviewer_bHUs · 2024-11-21
> > >
> > > Thank you for the detailed response, which addressed most of my concerns. I'm still a bit confused about STIC-1 and STIC-2. Taking floor planning as a concrete example. Suppose we are planning a 2-level building with 3 constraints, where TS_1="floor1 must have a coffee shop", TS_2="floor1 must have a reception desk", TP_1="every floor must have a washroom". The model generates "floor1: coffee shop, washroom; floor2: washroom". My understanding is that STIC-2 will be 0 for this problem because the model does not get floor1 correct. But I'm not sure whether STIC-1 = 1/2 because floor1 is incorrect but floor2 is correct, or STIC-1=2/3 because TS_1 and TP_1 are satisfied but TS_2 is not satisfied.

---

> ### Author Response · Authors · 2024-11-22
> **Further explanation of STIC-1 and STIC-2**
>
> Thank you for your careful reading and for delving into the discussion. I really appreciate your example; it's clearer and simpler than the one I provided. Allow me to use it to explain the concepts further.
>
> First, I've slightly modified your example from a 2-level building to a 3-level building to make the explanation clearer:
>
> Suppose we are planning a "3"-level building with 3 constraints:
> - **TS_1**: "floor1 must have a coffee shop"
> - **TS_2**: "floor1 must have a reception desk"
> - **TP**: `{TP_1, TP_2, TP_3}`, where each `TP_i` means "floor i must have a washroom"
>
> The model generates:
> > "floor1: coffee shop, washroom; floor2: washroom."
>
> In this scenario, my **check_set** is `{TS_1, TS_2, TP_1, TP_2, TP_3}`. It's important to note that `TP` actually applies to all three floors, meaning we need to evaluate each `TP_i` separately.
>
> In our actual evaluation, we strive to ensure that **TS**, **TR**, and **TP** have similar evaluation frequencies.
>
> With the current model output, the **completion rate (CR)** for the main task is **2/3**. Although the task requires outputs for 3 floors, the model only provided outputs for 2 floors.
>
> Now, for **STIC-1**, we consider how accurately the model has outputted information at the floor level. Since the model output only contains two floors, we evaluate the constraints for these two floors to see if they are fully met. For these two floors, the constraints are `TS_1, TS_2, TP_1, TP_2`, totaling 4 constraints. The model has correctly fulfilled 3 out of these 4 requirements, which means **STIC-1** is **3/4**.
>
> For **STIC-2**, we evaluate the entire **check_set**, which consists of `TS_1, TS_2, TP_1, TP_2, TP_3`. The model has fulfilled 3 out of these 5 requirements, so **STIC-2** equals **3/5**.
>
> The distinction between **STIC-1** and **STIC-2** allows us to identify the specific reasons for any drop in performance. It helps to determine whether the issue lies in the model's inability to follow instructions for a given output or whether it lacks a complete output in the first place. For example, in the case of a lower **STIC-2**, is the low score due to having some floor outputs that are incorrect, or is it because there is no complete output for the floors at all? In such cases, we can use **CR** and **STIC-1** together to further evaluate and make judgments.
>
> Once again, we greatly appreciate your comment and the opportunity to clarify the computation of our metrics. We will add this discussion in the appendix to improve the clarity of our paper. If this response addresses your concern, we hope you will consider raising the score. If there are further questions, we are happy to clarify further :)

---

> > ### Comment · Reviewer_bHUs · 2024-11-22
> >
> > Thanks for the further clarification. Assuming the discussion will be incorporated into the camera ready version, I raised my score.

---

### Official Review · Reviewer_hgz3 · 2024-11-03

**Soundness:** 2
**Presentation:** 3
**Contribution:** 2
**Rating:** 3
**Confidence:** 3

**Summary:**

The paper proposes a benchmark to evaluate models' strength in long-span generation tasks. It constructs synthetic examples from a fixed set of scenarios and templates in three modes. The resultant instruction measures models' abilities to faithfully follow position-specific, range-specific, and periodic instructions.

**Strengths:**

1. The paper is the first attempt at long-context generation, requiring models to generate a long text that follows a combination of specific instructions as opposed to just answering questions pertaining to long prompts.
2. The benchmark does uncover a setting that seems to be challenging to SOTA models.

**Weaknesses:**

1. The paper is very sparse on details regarding the evaluation of correctness. How is matching and parsing done w.r.t. the templates? A model could generate several outputs matching the criteria
2. The benchmark is limited to a few domains and scenarios, and the paper's contributions seem quite limited overall

**Questions:**

Would an IFEval [1] style setting where the outputs or attributes of the outputs could be verified definitively using code checks be a better option for such a benchmark? For long generations, getting the model outputs to match specific output patterns in the prompt is a challenge unto itself.

[1] https://arxiv.org/abs/2311.07911

---

> ### Author Response · Authors · 2024-11-18
> **Comment (1/2)**
>
> Thank you for your constructive review and valuable suggestions! Below, we provide a detailed response to your questions and comments. If any of our responses fail to sufficiently address your concerns, please inform us, and we will promptly follow up.
>
> **W1:Evaluation of Correctness**
>
> Thank you for your questions and for prompting us to clarify our evaluation process further. We realize our original explanation may have lacked detail, so we are pleased to provide a more comprehensive breakdown of the evaluation pipeline here, as also outlined in Section 2.5 of the paper.
>
> Our evaluation pipeline systematically assesses the ability of long-context LLMs to follow specific, complex instructions. The process can be summarized in three key steps:
>
> ### 1. Generation of Outputs from the Long-context LLM
>
> Given an input task (`T`) that describes a set of instructions, we prompt the LLM to generate detailed outputs. The output (`A`) comprises a list of descriptions, represented as: `A = {A1, A2, ..., An}`
>
>
> **Example: Given the prompt (ref Appendix SCENARIO)**
> > **Construct a skyscraper with 100 floors.** The floor assignments are detailed as follows:
> > - **Specific floor requirement:** Designate Floor 11 for a small art gallery.
> > - **Range floor requirement:** Allocate Floors 32 to 39 for corporate headquarters of a major company.
> > - ...
>
> The LLM generates a response describing each floor in detail, such as:
> > - Floor 1: ... Lobby ...
> > -   ...
> > - Floor 11: ... Small art gallery ...
> > -   ...
> > - Floor 32: ... Corporate headquarters ...
> > -   ...
> > - Floor n: ...
>
> ### 2. Extracting and Matching Relevant Floor Assignments (Check Set)
>
> From the initial input ("T"), we create a **check set** containing specific floor assignments to verify if the LLM correctly follows the instructions.
>
> For the example above, the check set includes:
> > - Floor 11: Small art gallery
> > - Floor 32: Corporate headquarters
> > - Floor 33: Corporate headquarters
> > - ...
>
> We then extract the relevant parts of the LLM output ("A") that correspond to the floor assignments described in the check set.
>
> ### 3. Evaluation Using Llama 3.1-8B instruction Model
>
> For each extracted pair, we use the Llama 3.1-8B model to evaluate whether the output ("Ai") for a given task segment ("Tsi") has correctly fulfilled the specified instruction.
>
> This evaluation task is framed as a simple **binary classification** problem, which aims to determine if the specific instruction was fulfilled ("yes" or "no"). The prompt used for this evaluation is as follows:
>
> **Evaluation Prompts**
> > - *Example 1*: XXXX **Answer:** Analysis + #*# Yes
> > - *Example 2*: XXXX **Answer:** Analysis + #*# No
> > - **Context:** Long-context model output: *"Floor 11: ... small art gallery ..."*
> > - **Instructions:** Does this context include 'small art gallery'?
> > - **Answer:** Please refer to the above example, provide your analysis, and respond with either #*# Yes or #*# No.
>
> Notably, this binary evaluation is straightforward. We manually labeled 300 data points, and the model's output matched human evaluations for all cases.
>
> By segmenting the long-generation task into smaller units and evaluating each one individually, our approach offers a thorough and systematic method to verify instruction adherence across the full sequence. This ensures that the LLM’s performance on each component of the task can be accurately and efficiently assessed.
> We hope this detailed explanation clarifies our approach, and we thank you for the opportunity to elaborate on our evaluation methodology.

---

> ### Author Response · Authors · 2024-11-18
> **Comment (2/2)**
>
> **W2:Benchmark Scope and Contributions**
>
> Thank you for your feedback regarding the scope of domains and scenarios in our benchmark. We believe that the key contribution of our paper lies not merely in the variety of domains but in establishing a systematic framework for evaluating instruction-following capabilities specifically tailored to long-context generation—a critical yet underexplored area in LLM research.
>
> Our benchmark addresses the essential challenge of instruction adherence over extended text generation, which is foundational to long-form tasks in real-world applications. To rigorously evaluate this capability, we designed four distinct scenarios with two length variations each, resulting in a total of 800 test samples and approximately 14,000 instructions. This setup is substantial and exceeds the size of many existing datasets focused on creative writing [1] and instruction-following [2,3]. Through this carefully structured benchmark, our work introduces a comprehensive, replicable framework that captures key aspects of long-context generation, making it possible to directly compare models on their ability to maintain instruction adherence across extended sequences.
>
> We recognize that our current benchmark does not yet encompass all potential real-world applications, and expanding to more diverse domains is indeed an important future direction. However, we are confident that this initial framework fills a critical gap by focusing on instruction-following in long-form text generation, establishing a strong foundation upon which future benchmarks can build. Additionally, we plan to broaden the benchmark to include narrative coherence, factual consistency, and creative tasks in subsequent iterations and will be introducing a public leaderboard to support ongoing evaluation and continuous model integration.
>
> We appreciate your feedback, which underscores valuable directions for further enhancement, and believe that our work makes a meaningful contribution by creating the first structured and scalable benchmark specifically for assessing long-context generation instruction-following capabilities in LLMs.
>
>
>
> References:
>
> [1] Comparison of Evaluation Metrics for Short Story Generation.
>
> [2] Instruction-following Evaluation for Large Language Models.
>
> [3] Infobench: Evaluating Instruction Following Ability in Large Language Models.
>
>
> **Q1: IFEval-style Setting and Code-based Verification**
>
> Thank you for this insightful suggestion. We have thoroughly reviewed IFEval, and we acknowledge that it effectively uses code checks to ensure correctness for certain attributes, such as output length and basic properties. This approach is beneficial for simpler, well-defined tasks where outputs can be evaluated through direct comparisons. However, IFEval has limitations when applied to more complex, nuanced characteristics like instruction-following accuracy, particularly for long-form generation tasks that require adherence to sophisticated requirements.
>
> As we mentioned in our response to W1, our approach addresses this gap by enabling evaluation of long-form outputs with a focus on verifying specific requirements from the prompts by LLMs. This focus is crucial for ensuring adherence to complex instructions, as it allows us to assess whether the model meets each component of the prompt accurately and consistently over extended text sequences.
>
> While we recognize the value of code-based checks, especially for tasks that require precise formatting or strict structural adherence, our current benchmark prioritizes the flexible evaluation necessary for long-form generation. That said, we are open to exploring how elements of IFEval or similar methodologies could complement our framework, particularly for tasks where structural constraints are more relevant.
>
> If there are additional aspects of IFEval or other methodologies you would like us to consider further, we would be more than happy to incorporate them into our analysis. We look forward to any further suggestions you may have.
>
> Reference:
>
> [1] IFEval: An Integrated Framework for Evaluating Instruction Following in LLMs.

---

> ### Author Response · Authors · 2024-11-22
>
> Dear Reviewer hgz3,
>
> Once again, thank you for your valuable feedback on our paper. We hope our clarifications and revisions have resolved the issues you highlighted. If there are any remaining questions or areas where further clarification would be helpful, we would be more than happy to address them promptly.
>
> As we are nearing the end of the rebuttal period, we kindly request you consider raising our paper's score if our updated responses have addressed your concerns.
>
> Thank you for your time and effort in reviewing our work.
>
> Best regards,
>
> Authors of Submission Number: 3588

---

> ### Author Response · Authors · 2024-11-25
> **Follow-Up on Rebuttal for Paper Submission 3588**
>
> Dear Reviewer hgz3,
>
> Thank you once again for your thoughtful feedback on our paper. We appreciate the time and effort you have invested in reviewing our work.
>
> As the Discussion Period is nearing its conclusion, we wanted to kindly follow up to ensure that you’ve had a chance to review our response to your comments. We hope that our rebuttal has addressed your concerns satisfactorily. If there are any additional points or further clarifications required, we would be happy to provide them promptly.
>
> Additionally, if our response has sufficiently addressed your concerns, we would greatly appreciate if you could consider reflecting this in your evaluation, including revisiting your score if appropriate.
>
> Thank you for your time and understanding. We greatly value your feedback and look forward to hearing from you.
>
> Cheers,
>
> Authors of Paper Submission 3588

---

> ### Comment · Reviewer_hgz3 · 2024-11-26
> **Thank You For The Response**
>
> Thank you for adding the response parsing and evaluation details in the response and the paper. I will maintain my score for the following two reasons (which are somewhat related):
>
> 1. LLM-as-a-judge style evaluations are alright for domains such as long context creative writing in benchmarks such as LongBench-Write [1]. However, in evaluating faithfulness in following instructions over long generations, an LLM-based evaluator leaves much to be desired, as it is brittle to generation parameters and the generation/formatting styles of the model under test.
>
> 2. I am not convinced that a long context benchmark that isn't grounded in some deterministically verifiable ground truth (like IFEval [2]) is consigned to be simplistic. In fact, in the math and code domain (which most recent LM releases are trained on), we have observed several challenging benchmarks in the last few months, which do a very good job of testing long context understanding and generation while also evaluating based on execution or answer correctness [3,4,5,6].
>
> -----------------------------------------------------------------------------------------------------------------------
>
> [1] 	Yushi Bai, Jiajie Zhang, Xin Lv, Linzhi Zheng, Siqi Zhu, Lei Hou, Yuxiao Dong, Jie Tang, Juanzi Li:
> LongWriter: Unleashing 10,000+ Word Generation from Long Context LLMs. CoRR abs/2408.07055 (2024)
>
> [2] Jeffrey Zhou, Tianjian Lu, Swaroop Mishra, Siddhartha Brahma, Sujoy Basu, Yi Luan, Denny Zhou, Le Hou: Instruction-Following Evaluation for Large Language Models. CoRR abs/2311.07911 (2023)
>
> [3] Carlos E. Jimenez, John Yang, Alexander Wettig, Shunyu Yao, Kexin Pei, Ofir Press, Karthik R. Narasimhan: SWE-bench: Can Language Models Resolve Real-world Github Issues? ICLR 2024
> 2023
>
> [4] Nam Le Hai, Dung Manh Nguyen, Nghi D. Q. Bui: REPOEXEC: Evaluate Code Generation with a Repository-Level Executable Benchmark. CoRR abs/2406.11927 (2024)
>
> [5] Egor Bogomolov, Aleksandra Eliseeva, Timur Galimzyanov, Evgeniy Glukhov, Anton Shapkin, Maria Tigina, Yaroslav Golubev, Alexander Kovrigin, Arie van Deursen, Maliheh Izadi, Timofey Bryksin:
> Long Code Arena: a Set of Benchmarks for Long-Context Code Models. CoRR abs/2406.11612 (2024)
>
> [6] Lei Wang, Shan Dong, Yuhui Xu, Hanze Dong, Yalu Wang, Amrita Saha, Ee-Peng Lim, Caiming Xiong, Doyen Sahoo: MathHay: An Automated Benchmark for Long-Context Mathematical Reasoning in LLMs. CoRR abs/2410.04698 (2024)

---

> ### Author Response · Authors · 2024-11-26
> **Comment-2 (1/2)**
>
> Thank you for your feedback! Please allow me to elaborate further:
>
> ***R1: Response to LLM-as-Jugde and LongBench-Write***
>
> We recognize the challenges associated with this approach, particularly its sensitivity to generation parameters and formatting styles. However, our work distinguishes itself by addressing these limitations through a structured and systematic evaluation process tailored specifically for long-form instruction adherence.
>
> Our paper introduces a novel benchmark specifically designed to evaluate the ability of large language models (LLMs) to follow detailed instructions over extended outputs. Unlike creative writing tasks, such as those in LongWrite [1], our benchmark emphasizes instruction adherence in practical, structured tasks with significantly longer outputs (average lengths nearly **ten times** greater). This focus tackles the unique challenges of long-context generation, including consistency, memory retention, and instruction fidelity, which are foundational for advancing LLM capabilities in real-world applications.
>
> To ensure robust evaluation, we adopt a segment-based approach. Instead of relying solely on an **LLM-as-a-judge** paradigm, we split long outputs into manageable sub-segments (generally **100–200 words**) and systematically verify whether each segment meets the given instructions. For example, if the instruction specifies that “_Floor 34 must include a coffee shop_”, we evaluate whether this requirement is explicitly satisfied in the corresponding segment (Appendix C & Comment(1/2)). This method minimizes dependencies on generation parameters or formatting styles and provides a transparent, reproducible evaluation framework.
>
> While LongWrite uses **LLM-as-a-judge**, which has faced critiques for its brittleness and susceptibility to formatting (see https://openreview.net/forum?id=kQ5s9Yh0WI), our evaluation methodology is tailored to address these limitations. Furthermore, Long Write’s average output length of **2,772** words represents a fundamentally different scope, focusing on creative writing rather than instruction adherence over ultra-long contexts.
>
> By emphasizing long-context instruction-following, we provide a critical framework for assessing a capability that underpins many real-world applications. This distinction sets our benchmark apart as a foundational contribution to the evaluation of long-context LLM outputs. We hope this clarifies the contributions and robustness of our evaluation.
>
> ***R2: Response to Long-Context Benchmark and Deterministic Verifiability***
>
> Below, we summarize the average token counts for the datasets and benchmarks you referenced:
>
> **IFEval [2]**:  **344** tokens.
>
> **SWE-BENCH[3]**: **120** words (Table 1)
>
> **REPOEXEC [4]**: **78.46** tokens (Table 2).
>
>
> **Long Code Arena: A Set of Benchmarks for Long-Context Code Models [5]**:
> - Library-based Code Generation: generates a **single** file (largest task).
> - Project-Level Code Completion: typically generates **single-line** code.
> - Average file size: **32.5** lines (Table 8).
>
> **MATHHAY: An Automated Benchmark for Long-Context Mathematical Reasoning in LLMs [6]**: Focuses on processing long inputs rather than generating long outputs, combining **information-seeking** and **mathematical reasoning** tasks.
>
> While these works demonstrate strong capabilities in long-context understanding, they **do not** primarily address **long-context generation**, which we define as producing outputs that exceed 4K tokens, with some tasks in our benchmark requiring significantly longer outputs. Our focus is distinct in evaluating the challenges of maintaining coherence, adhering to instructions, and handling memory over ultra-long text outputs.
>
> This makes our work among the first to systematically benchmark long-context generation. Existing benchmarks, such as IFEval and REPOEXEC, primarily focus on short or moderately long outputs with deterministic evaluation criteria, such as correctness checks for code or mathematical reasoning. These approaches, while valuable, are not directly applicable to the open-ended and complex nature of long-context generation tasks (No ground truth), which require a broader and more flexible evaluation framework.
>
> Given the reasons mentioned above, we believe that using LLMs for fragment evaluation to determine whether an instruction has been completed will not lead to the issues you described. As you noted, existing works like Longwrite are limited to using LLMs for quality evaluation. In contrast, our approach provides greater explainability in evaluation, allowing us to directly identify model output errors instead of relying on vague metrics such as creativity.
>
> Let me know if this works for you or if you'd like further adjustments!

---

> ### Author Response · Authors · 2024-11-26
> **Comment-2 (2/2)**
>
> We would like to re-iterate our contribution in this work:
>
> **Instruction-following Over Extended Outputs**:
> We tackle the problem of models adhering to detailed and complex instructions across outputs far longer than those evaluated in previous benchmarks. This requires addressing issues such as instruction retention, content diversity, and semantic correctness at scale.
>
> **Foundational Benchmark for Long-context Generation**:
> While prior works have laid the groundwork for long-context understanding, we focus specifically on generation challenges, providing a structured framework that enables rigorous evaluation of these capabilities.
>
> By addressing these unique challenges, our work sets the stage for future research in long-context generation. It provides a foundational benchmark for the community to systematically assess model performance in scenarios that demand both scale and complexity.
>
> We hope this clarifies the significance of our contributions compared to the benchmarks you referenced.
>
> ---
>
> [1] Yushi Bai, Jiajie Zhang, Xin Lv, Linzhi Zheng, Siqi Zhu, Lei Hou, Yuxiao Dong, Jie Tang, Juanzi Li: LongWriter: Unleashing 10,000+ Word Generation from Long Context LLMs. CoRR abs/2408.07055 (2024)
>
> [2] Jeffrey Zhou, Tianjian Lu, Swaroop Mishra, Siddhartha Brahma, Sujoy Basu, Yi Luan, Denny Zhou, Le Hou: Instruction-Following Evaluation for Large Language Models. CoRR abs/2311.07911 (2023)
>
> [3] Carlos E. Jimenez, John Yang, Alexander Wettig, Shunyu Yao, Kexin Pei, Ofir Press, Karthik R. Narasimhan: SWE-bench: Can Language Models Resolve Real-world Github Issues? ICLR 2024
>
> [4] Nam Le Hai, Dung Manh Nguyen, Nghi D. Q. Bui: REPOEXEC: Evaluate Code Generation with a Repository-Level Executable Benchmark. CoRR abs/2406.11927 (2024)
>
> [5] Egor Bogomolov, Aleksandra Eliseeva, Timur Galimzyanov, Evgeniy Glukhov, Anton Shapkin, Maria Tigina, Yaroslav Golubev, Alexander Kovrigin, Arie van Deursen, Maliheh Izadi, Timofey Bryksin: Long Code Arena: a Set of Benchmarks for Long-Context Code Models. CoRR abs/2406.11612 (2024)
>
> [6] Lei Wang, Shan Dong, Yuhui Xu, Hanze Dong, Yalu Wang, Amrita Saha, Ee-Peng Lim, Caiming Xiong, Doyen Sahoo: MathHay: An Automated Benchmark for Long-Context Mathematical Reasoning in LLMs. CoRR abs/2410.04698 (2024)

---

### Official Review · Reviewer_Nybu · 2024-11-04

**Soundness:** 3
**Presentation:** 3
**Contribution:** 2
**Rating:** 5
**Confidence:** 4

**Summary:**

The paper introduces LongGenBench, a benchmark for evaluating large language models' (LLMs) ability to generate long-form text while adhering to complex instructions. It features tasks in four scenarios with different instruction types and lengths (16K and 32K tokens). The study finds that even advanced LLMs struggle with long-form generation, particularly as text length increases, highlighting the need for improvements in model architecture and training.

**Strengths:**

1. Interesting task design, which can evaluate the long text generation ability of large models from a certain perspective
2. The paper is well written.

**Weaknesses:**

1. The types of task scenarios are relatively limited, and it is impossible to comprehensively evaluate the long text generation capabilities of large models.
2. The evaluation metrics seem to be customized according to the scenario.
3. Limited number of models evaluated

**Questions:**

1. It seems that the evaluation metrics are designed for these scenarios. If there are new scenario tasks, do we need to update the evaluation metrics?
2. What other aspects of long text generation with LLM do you think need to be evaluated? It seems that your evaluation is more oriented towards some planning tasks or well-structured text. Is it difficult to evaluate the creation task? For example, it is difficult to design evaluation metrics for novel writing?

---

> ### Author Response · Authors · 2024-11-18
> **Comment (1/2)**
>
> Thank you for your constructive review and valuable suggestions! Below, we provide a detailed response to your questions and comments. If any of our responses fail to sufficiently address your concerns, please inform us, and we will promptly follow up.
>
> **W1: Limited Task Scenarios**
>
> Thank you for your feedback regarding the variety of task scenarios in our benchmark. In this initial study, we intentionally narrowed the scope to focus specifically on evaluating instruction-following capabilities within the realm of long-text generation. Our goal was to address a critical component of long-text generation systems that serves as a foundation for more complex and comprehensive tasks.
>
> Evaluating the full performance of long-text generation in diverse applications is indeed a significant challenge that requires ongoing and extensive research efforts. Our contribution to this complex field is to spotlight instruction adherence as a key aspect, which we believe is pivotal for the success of long-text generation systems. By focusing on this foundational element, we aim to provide a structured approach that can serve as a stepping stone for more comprehensive evaluations.
>
> For future iterations, we plan to expand the benchmark to cover a broader range of scenarios, including tasks that test narrative coherence, factual consistency, and reasoning. These additions will allow us to capture more advanced aspects of long-text generation and increase the benchmark’s applicability to a wider array of real-world applications.
> We appreciate your feedback, as it highlights the importance of broadening our benchmark to evaluate additional capabilities, and we look forward to making these expansions in future releases.
>
> **W2/Q1: Scenario-specific Evaluation Metrics**
>
> Thank you for your observations regarding the scenario-specific nature of our evaluation metrics. We believe that there is a strong intrinsic connection between the metric and the task it evaluates. Should the proposed metric effectively assess model performance for these tasks, its suitability is affirmed, like FUAR metric to Continual knowledge learning[1], TRUST-SCORE metric to trustworthiness RAG[2].
>
> In this benchmark, we designed metrics specifically to assess instruction-following capabilities in long-text generation, similar to how the FUAR and TRUST-SCORE. For the present scenarios, our metrics effectively measure compliance with provided instructions and offer a clear view of how well models handle extended, instruction-based tasks.
>
> **Flexibility for Future Scenarios**
>
> While our current metrics are customized to evaluate instruction-following, we recognize that additional metrics would be needed to capture broader dimensions of long-text generation, such as narrative coherence, fluency, and factual consistency (please ref Section 4 ANALYSIS AND LIMITATIONS). Should we expand the benchmark to include tasks with these aspects, we would introduce complementary metrics suited to those evaluation goals. For example, creative storytelling tasks may benefit from metrics focusing on coherence and thematic consistency, while technical report generation could require metrics assessing factual accuracy and data integrity.
>
> We appreciate your feedback, which will guide us in adapting our metrics to accommodate additional tasks and evaluation needs in future releases.
>
> References:
>
> [1] Towards Continual Knowledge Learning of Language Models.
>
> [2] Measuring and Enhancing Trustworthiness of LLMs in RAG through Grounded Attributions and Learning to Refuse.

---

> ### Author Response · Authors · 2024-11-18
> **Comment (2/2)**
>
> **W3: Limited Number of Models Evaluated**
>
> Thank you for your constructive feedback. In this work, we conducted an extensive evaluation that included models of various sizes and sources, encompassing both open-source and close-source options. To address your suggestion, we have incorporated additional models, including Phi-3-mini-instruct, Phi-3.5-MOE-instruct [1], FILM-7B [2], and Mamba-2.8B [3] into our experiments, as shown in the table below:
>
> |            |           |  16K           |       |      |      |      |          32K           |       |      |      |      |
> |-----------------------|--------------|-------|--------|--------|--------|--------------|-------|--------|--------|--------|--------|
> |   Model           | Claim length | CR    | STIC-1 | STIC-2 | length | wAvg | CR    | STIC-1 | STIC-2 | length | wAvg |
> | Phi-3-mini-instruct   | 128K         | 22.9% | 27.6%  | 5.4%   | 4165   | 1.2% | 7.4%  | 46.9%  | 2.4%   | 2613   | 0.2% |
> | mamba-2.8b            | 2K           | 11.3% | 23.8%  | 2.1%   | 902    | 0.2% | 5.6%  | 29.8%  | 1.6%   | 864    | 0.1% |
> | FILM-7B               | 32K          | 36.0% | 9.9%   | 3.9%   | 6280   | 1.4% | 37.4% | 30.9%  | 10.9%  | 13775  | 4.1% |
> | Phi-3.5-MoE-instruct  | 128K         | 26.9% | 46.4%  | 11.3%  | 5430   | 3.0% | 7.4%  | 62.9%  | 6.0%   | 6633   | 0.4% |
>
> All of the above models perform poorly, which we attribute to the lack of SFT training in the long output case. On the other hand, since most of the APIs for closed-source models can only support outputs up to 4K, we are not able to perform a more comprehensive evaluation of closed-source models.
> In addition, we plan to create a public leaderboard, where we will continuously integrate more models to enable broader comparisons and ensure transparency. This leaderboard will allow researchers to track model performance on long-text generation tasks, making it easier to evaluate newer models as they emerge.
> If there are specific models you would like us to include, please let us know, and we would be more than happy to add them to our analysis.
>
> References:
>
> [1] Phi-3 Technical Report: A Highly Capable Language Model Locally on Your Phone.
>
> [2] Make Your LLM Fully Utilize the Context.
>
> [3] Mamba: Linear-Time Sequence Modeling with Selective State Spaces.
>
>
>
>
> **Q2: Other Aspects of Long Text Generation with LLMs**
>
> Thank you for your insightful observation regarding the limitations of our current approach. As noted in Section 4 (ANALYSIS AND LIMITATIONS) of our paper, we recognize that our current research focuses primarily on evaluating instruction-following capabilities, while a more comprehensive analysis of content coherence and rationality remains an area for future investigation.
>
> Our work represents the first comprehensive evaluation focused on long-context generation, providing a structured and replicable foundation for assessing large language models (LLMs) in this critical area. By systematically evaluating instruction-following and planning capabilities in extended contexts, our benchmark addresses foundational skills that are essential for any long-form generation task. This approach offers the community a baseline for understanding model performance in long-context settings, allowing future research to build on this framework as evaluation methods and LLM capabilities evolve.
>
> We agree that evaluating other aspects of long-text generation, such as creative tasks like novel writing, presents unique challenges that go beyond structured or planning-based tasks. Designing effective evaluation metrics for open-ended creative content is inherently complex, as it requires balancing subjective qualities like narrative coherence, creativity, and reader engagement—elements that are more difficult to quantify objectively.
>
> We appreciate your feedback, which highlights valuable directions for expanding our benchmark to cover more diverse aspects of long-text generation. By incorporating these additional dimensions in future iterations, we aim to provide a more comprehensive assessment of LLM capabilities in generating long, complex, and creative content.

---

> ### Author Response · Authors · 2024-11-22
>
> Dear Reviewer Nybu,
>
> Once again, thank you for your valuable feedback on our paper. We hope our clarifications and revisions have resolved the issues you highlighted. If there are any remaining questions or areas where further clarification would be helpful, we would be more than happy to address them promptly.
>
> As we are nearing the end of the rebuttal period, we kindly request you consider raising our paper's score if our updated responses have addressed your concerns.
>
> Thank you for your time and effort in reviewing our work.
>
> Best regards,
>
> Authors of Submission Number: 3588

---

> ### Author Response · Authors · 2024-11-25
> **Follow-Up on Rebuttal for Paper Submission 3588**
>
> Dear Reviewer Nybu,
>
> Thank you once again for your thoughtful feedback on our paper. We appreciate the time and effort you have invested in reviewing our work.
>
> As the Discussion Period is nearing its conclusion, we wanted to kindly follow up to ensure that you’ve had a chance to review our response to your comments. We hope that our rebuttal has addressed your concerns satisfactorily. If there are any additional points or further clarifications required, we would be happy to provide them promptly.
>
> Additionally, if our response has sufficiently addressed your concerns, we would greatly appreciate if you could consider reflecting this in your evaluation, including revisiting your score if appropriate.
>
> Thank you for your time and understanding. We greatly value your feedback and look forward to hearing from you.
>
> Cheers,
>
> Authors of Paper Submission 3588

---

> > ### Comment · Reviewer_Nybu · 2024-11-26
> >
> > Thank you very much for your detailed explanation, which has resolved all my concerns. I will maintain my current rating.

---

> > > ### Author Response · Authors · 2024-11-26
> > >
> > > Thank you very much for your thoughtful engagement and for acknowledging that our detailed explanations have addressed all your concerns. We deeply appreciate the time and effort you have dedicated to reviewing our work and providing constructive feedback, which has helped improve the clarity and quality of our paper.
> > >
> > > Given that we have resolved the issues you previously raised, we kindly ask if you would consider reevaluating your current score of 5. Our work presents a novel contribution to the evaluation of long-context generation, tackling a critical gap in existing benchmarks with a focus on ultra-long outputs and instruction adherence. We believe that the unique challenges addressed in our benchmark and its potential impact on advancing long-context language model research align well with the standards for acceptance.
> > >
> > > We fully respect your decision, but any reconsideration would mean a great deal to us, as we strive to have this work recognized and shared with the community. Thank you again for your invaluable feedback and support.

---

> > > ### Author Response · Authors · 2024-11-27
> > >
> > > Once again, thank you for your valuable feedback and for acknowledging that our response has addressed all your concerns. We truly appreciate the time and effort you have taken to review our submission thoroughly.
> > >
> > > Given that we have addressed all the concerns raised, we kindly request that you reconsider your score to reflect the improvements made to our submission better. We believe that a higher score would align more closely with the current state of the paper and its potential contribution to the field.
> > >
> > > Should you have further concerns, we are happy to address them and clarify further.

---

> ### Author Response · Authors · 2024-11-29
>
> We would like to take this opportunity to further clarify some of the points related to W1, W2, and Q1. Your insightful comments on benchmarking demonstrate a deep understanding of this research area, and we are grateful for your thoughtful engagement.
>
> We believe you may be familiar with the work on abstract visual reasoning using Tangram shapes. That study introduced a dataset constructed with Tangram pieces to evaluate the abstract visual reasoning capabilities of VLMs. Alongside this, the authors proposed metrics such as Shape Naming Divergence (SND), Part Naming Divergence (PND), and Part Segmentation Agreement (PSA), which were specifically designed for their KILOGRAM benchmark. While the benchmark did not encompass all aspects of abstract visual reasoning, the work’s novelty and contributions were significant enough to earn it the **Best Paper** award at EMNLP 2022.
>
> Similarly, our work introduces a benchmark and evaluation metrics tailored to a **specific instruction-following** task within the broader domain of long-form text generation. While our metrics are designed with our benchmark in mind, their primary purpose is to provide **accurate performance assessment for this emerging subfield**. Generalizability, while important, is a secondary consideration at this stage.
>
> As **one of the initial** works in this subfield, we kindly ask that you consider our benchmark with an **open and inclusive** **perspective**, recognizing its potential to lay a foundation for further advancements in long-context text generation. We deeply appreciate your understanding, your insightful feedback, and your consideration.
>
> ps: happy thanksgiving everyone!

---

> > ### Comment · Reviewer_Nybu · 2024-12-02
> >
> > Thank you for the additional results again. After carefully reviewing the updates, I have decided to maintain my score.

---

### Official Review · Reviewer_p7FN · 2024-11-04

**Soundness:** 3
**Presentation:** 2
**Contribution:** 3
**Rating:** 8
**Confidence:** 3

**Summary:**

Existing LLMs' long-context benchmarks rely on understanding tasks. This paper proposes a benchmark targeting specifically on long-context *generation* ability of LLMs. The authors design 4 long-context instruction-following tasks, up to 16 or 32K tokens: (1) Writing Diary for a year; (2) Wrting menu for a year; (3) Design a 100/300-floor skyscraper; (4) Plan an urban layout. As a result, all existing LLMs don't work well on these tasks of the benchmark.

**Strengths:**

* This paper is overall clear and easy to understand.
* This proposed evaluation is novel, and the generation ability it benchmarks is not covered by previous metrics.

**Weaknesses:**

* Some of the details are possibly missing or hard to get by readers -- see "Questions".
* In the proposed benchmark, the way to form long content is to pile short answers to many sub-queries, while the sub-tasks are actually independent, to a large extent. For example, given all the demands on one-year dairies, it should be easy for LLMs to write a diary if it is assigned a specific day of the year, while this benchmark just require the LLM generate 365 diaries all at once. In this case, the challenge of this benchmark might majorly be forgetting the instruction under the influence of generated content, instead of keeping the conherence and content interaction among generated long content. That latter should be the one mostly desired by the community.

**Questions:**

* How do you split the long generation and match them to all the subtask instructions?
* How do you check if every sub-instruction is satisfied? is it by prompting another LLM or by word matching, etc.?
* What's the significant difference between STIC-1 and STIC-2? Looks they just have difference denominators. I don't quite get it although there is a paragraph in Sec. 2.4 as below for this. Is there any specific case where an LLM can get low STIC-1 while high STIC-2, or the other way around?

> STIC-1 is primarily concerned with the completion rate of instructions that result in sub-scenarios,
focusing on whether instructions are correctly executed. In contrast, STIC-2 assesses the overall completion of the specific instruction task, including the presence of sub-scenarios and their completion
status.

---

> ### Author Response · Authors · 2024-11-18
> **Comment (1/3)**
>
> Thank you for your constructive review and valuable suggestions! Below, we provide a detailed response to your questions and comments. If any of our responses fail to sufficiently address your concerns, please inform us, and we will promptly follow up.
>
> **W2: Structure and Independence of Sub-tasks in Long Content**
>
> Thank you for your insightful comments regarding coherence and content interaction in long-form text generation. We recognize that coherence and interaction are essential for the community’s broader goals in long-text generation research. In our paper, we clarify that our current focus is not primarily on evaluating coherence across long outputs but rather on exploring the foundational aspects of long-text generation—specifically, the model’s ability to follow complex instructions over an extended sequence.
>
> Our benchmark is designed to establish a robust baseline for instruction-following capabilities, which we believe is a necessary precursor to effectively handling the more nuanced demands of content coherence and interaction in long-form generation. This focus is similar to the evolution of benchmarks like NIAH, which initially assessed long-context retrieval, leading to more advanced frameworks like RULER and NeedleBench that require deeper understanding and nuanced content interaction.
>
> In tasks like diary generation or skyscraper design, while individual entries may seem independent, the model must follow overarching instructions consistently over time, a quality essential for applications involving long-context generation. These tasks test whether models can remember and adhere to periodic and complex directives over extended sequences without "drifting" or forgetting instructions.
>
> While we acknowledge that content coherence and interactivity are critical dimensions for future work, our benchmark aims to test foundational capabilities first. As the benchmark evolves, we are committed to introducing tasks that address coherence and interdependent content more thoroughly, which will enhance the benchmark’s comprehensiveness. We look forward to advancing our work in this direction and appreciate your feedback, which will help inform our future iterations.

---

> ### Author Response · Authors · 2024-11-18
> **Comment (2/3)**
>
> **Q1/Q2: Sub-task Splitting and Instruction Satisfaction Evaluation**
>
> Thank you for your questions and for prompting us to clarify our evaluation process further. We realize our original explanation may have lacked detail, so we are pleased to provide a more comprehensive breakdown of the evaluation pipeline here, as also outlined in Section 2.5 of the paper.
>
> Our evaluation pipeline systematically assesses the ability of long-context LLMs to follow specific, complex instructions. The process can be summarized in three key steps:
>
> ### 1. Generation of Outputs from the Long-context LLM
>
> Given an input task (`T`) that describes a set of instructions, we prompt the LLM to generate detailed outputs. The output (`A`) comprises a list of descriptions, represented as: `A = {A1, A2, ..., An}`
>
>
> **Example: Given the prompt (ref Appendix SCENARIO)**
> > **Construct a skyscraper with 100 floors.** The floor assignments are detailed as follows:
> > - **Specific floor requirement:** Designate Floor 11 for a small art gallery.
> > - **Range floor requirement:** Allocate Floors 32 to 39 for corporate headquarters of a major company.
> > - ...
>
> The LLM generates a response describing each floor in detail, such as:
> > - Floor 1: ... Lobby ...
> > -   ...
> > - Floor 11: ... Small art gallery ...
> > -   ...
> > - Floor 32: ... Corporate headquarters ...
> > -   ...
> > - Floor n: ...
>
> ### 2. Extracting and Matching Relevant Floor Assignments (Check Set)
>
> From the initial input ("T"), we create a **check set** containing specific floor assignments to verify if the LLM correctly follows the instructions.
>
> For the example above, the check set includes:
> > - Floor 11: Small art gallery
> > - Floor 32: Corporate headquarters
> > - Floor 33: Corporate headquarters
> > - ...
>
> We then extract the relevant parts of the LLM output ("A") that correspond to the floor assignments described in the check set.
>
> ### 3. Evaluation Using Llama 3.1-8B instruction Model
>
> For each extracted pair, we use the Llama 3.1-8B model to evaluate whether the output ("Ai") for a given task segment ("Tsi") has correctly fulfilled the specified instruction.
>
> This evaluation task is framed as a simple **binary classification** problem, which aims to determine if the specific instruction was fulfilled ("yes" or "no"). The prompt used for this evaluation is as follows:
>
> **Evaluation Prompts**
> > - *Example 1*: XXXX **Answer:** Analysis + #*# Yes
> > - *Example 2*: XXXX **Answer:** Analysis + #*# No
> > - **Context:** Long-context model output: *"Floor 11: ... small art gallery ..."*
> > - **Instructions:** Does this context include 'small art gallery'?
> > - **Answer:** Please refer to the above example, provide your analysis, and respond with either #*# Yes or #*# No.
>
> Notably, this binary evaluation is straightforward. We manually labeled 300 data points, and the model's output matched human evaluations for all cases.
>
> By segmenting the long-generation task into smaller units and evaluating each one individually, our approach offers a thorough and systematic method to verify instruction adherence across the full sequence. This ensures that the LLM’s performance on each component of the task can be accurately and efficiently assessed.
> We hope this detailed explanation clarifies our approach, and we thank you for the opportunity to elaborate on our evaluation methodology.

---

> > ### Author Response · Authors · 2024-11-18
> > **Comment (3/3)**
> >
> > **Q3: Differences between STIC-1 and STIC-2**
> >
> > Thank you for your insightful comments regarding the need for an example to illustrate the differences between STIC-1 and STIC-2. We appreciate your feedback and have included a comparative example in the revised manuscript, specifically referencing results from Table 3 of our experiments, which compare LLaMA3.1-8B and Qwen2 under the short-version setting.
> > STIC-1 and STIC-2 are designed to evaluate instruction adherence at different levels of granularity:
> >
> > - **STIC-1** measures the average success rate of individual instructions within the actual completed portion of a task. For instance, STIC-1 evaluates the correctness of each generated output segment relative to the portion of the task completed, without penalizing for incomplete task segments. This allows STIC-1 to reflect the model’s consistency in following instructions within its generated content.
> > - **STIC-2**, in contrast, provides a comprehensive assessment of output completeness. This metric evaluates a task as a whole, counting it as successful only if all specified instructions across the full task length are followed correctly. STIC-2 thus captures the model’s ability to handle long and complex tasks comprehensively, without any partial completion.
> >
> > **Example Comparison**
> >
> > The table below compares LLaMA3.1-8B and Qwen2 to illustrate how these metrics diverge:
> >
> > | Model         | Length | CR    | STIC-1 | STIC-2 |
> > |--------|--------|-------|--------|--------|
> > | LLaMA3.1-8B   | 128K   | 93.5% | 23.4%  | 22.0%  |
> > | Qwen2-7B      | 128K   | 60.0% | 27.9%  | 16.1%  |
> >
> > In this case, Qwen2 achieves a higher STIC-1 score than LLaMA3.1-8B but a lower STIC-2 score. This difference arises from the models’ varying Completion Rates (CR). Qwen2 typically achieves a 60% completion rate, akin to completing approximately 60 floors of a 100-story skyscraper design task, while LLaMA3.1-8B completes closer to 93 floors.
> >
> > For **STIC-1**, Qwen2 scores higher since its evaluation is based only on the 60 floors it successfully generates, compared to LLaMA3.1-8B’s 93 floors. STIC-1 does not penalize Qwen2 for the missing floors, focusing instead on the instruction adherence within the portion generated. In contrast, **STIC-2** evaluates the completeness of the entire task; since Qwen2 does not generate the remaining 40 floors, its STIC-2 score is negatively impacted due to this incomplete output.
> >
> > We trust that this explanation clarifies the distinctions between STIC-1 and STIC-2, and we thank you for the opportunity to expand on these metrics in our revised manuscript.

---

> ### Author Response · Authors · 2024-11-22
>
> Dear Reviewer p7FN,
>
> Once again, thank you for your valuable feedback on our paper. We hope our clarifications and revisions have resolved the issues you highlighted. If there are any remaining questions or areas where further clarification would be helpful, we would be more than happy to address them promptly.
>
> As we are nearing the end of the rebuttal period, we kindly request you consider raising our paper's score if our updated responses have addressed your concerns.
>
> Thank you for your time and effort in reviewing our work.
>
> Best regards,
>
> Authors of Submission Number: 3588

---

> ### Author Response · Authors · 2024-11-25
> **Follow-Up on Rebuttal for Paper Submission 3588**
>
> Dear Reviewer p7FN,
>
> Thank you once again for your thoughtful feedback on our paper. We appreciate the time and effort you have invested in reviewing our work.
>
> As the Discussion Period is nearing its conclusion, we wanted to kindly follow up to ensure that you’ve had a chance to review our response to your comments. We hope that our rebuttal has addressed your concerns satisfactorily. If there are any additional points or further clarifications required, we would be happy to provide them promptly.
>
> Additionally, if our response has sufficiently addressed your concerns, we would greatly appreciate if you could consider reflecting this in your evaluation, including revisiting your score if appropriate.
>
> Thank you for your time and understanding. We greatly value your feedback and look forward to hearing from you.
>
> Cheers,
>
> Authors of Paper Submission 3588

---

> > ### Comment · Reviewer_p7FN · 2024-11-27
> >
> > Thanks to the authors for their clarifications! I feel my questions have been well addressed. Given the improved clarity and readability of the revised paper, I have decided to raise my score to 8 :)

---

> ### Author Response · Authors · 2024-11-27
>
> Many thanks four reviewing our paper and raising the score! We greatly appreciated your constructive review, which improved our paper!

---

### Official Review · Reviewer_dQFM · 2024-11-06

**Soundness:** 3
**Presentation:** 3
**Contribution:** 2
**Rating:** 8
**Confidence:** 3

**Summary:**

This paper introduces LongGenBench, a benchmark for measuring LLMs' capacities, especially their long-context abilities, by generating long-form context from 16k to 32k with rather complex instructions. This new dataset departs from traditional benchmarks aiming at decoded length in four different scenarios. Preliminary evaluations are done with main streamed LLMs.

**Strengths:**

- First benchmark focusing on long-form generation during the test time
- The evaluation combines both complexities of evaluation prompts and different scenarios
- First batch of results on 10 mainstreamed LLMs
- The paper is easy to follow

**Weaknesses:**

- I am a little bit distracted from the main takeaways from the experimental studies, and not so convinced with failure cases. See question 1

I have other minor concerns regarding the experiment setup

- There has been much research showing that the prompt format matters, what's your thought?

- Reasoning tasks are not well involved, as o1 seems to argue that longer decoded length is helpful with reasoning complex tasks, in your benchmark, you might want to add an axis of reasoning ability clearly or have some analysis around this topic?

- Most of the evaluation focused on existing transformer-based architecture, but models are presenting SOTA results, for example, mamba-based models. Are those models, with good inference time complexity, good at benchmark, or if not, why?

**Questions:**

- question 1: can you present a bit more concise takeaways from your benchmark, to me I feel like I was reading a lot of pieces, and no surprising results to me either. It might be good to have some failure cases to support your point

-  question 2: when you evaluate the prompt complexity, how do you choose the prompt formats?

-  question 3: Do you think complex prompts and instructions might need manyshots?

-  question 4: Do you think you can add some reasoning axes to your benchmark?

- question 5: maybe consider adding some long-context recent models with SOTA results, not only looking at model parameter counts but also architectural differences.

---

> ### Author Response · Authors · 2024-11-18
> **Comment (1/3)**
>
> Thank you for your constructive review and valuable suggestions! Below, we provide a detailed response to your questions and comments. If any of our responses fail to sufficiently address your concerns, please inform us, and we will promptly follow up.
>
> **W1/Q1: Main Takeaways and Failure Cases**
>
> Thank you for your feedback and for highlighting areas where additional clarity on takeaways and failure cases could enhance our paper. In response, we have taken several steps to provide a clearer overview of model performance and failure analysis.
> To further illustrate model limitations, we have expanded our error analysis in Appendix F (Original paper error analysis in appendix C), where we provide detailed examples of failure cases. Here, we illustrate issues encountered in the **Skyscraper Design task**, specifically highlighting where models struggled to consistently follow instruction requirements over extended sequences. Below is a concise breakdown of the errors in this example:
>
> ### Skyscraper Design Example
>
> **Objective**: Construct a skyscraper with 100 floors. The floor assignments are detailed as follows:
>
> - **Specific floor requirements**: Designate Floor 11 for a small art gallery.
> - **Range floor requirements**: Allocate Floors 32 to 39 for corporate headquarters of a major company.
> - **Periodic floor requirements**: Include a sky garden every 15 floors, starting from Floor 30.
>
> The output includes excerpts of floor descriptions and the corresponding correctness evaluation marked with either a green check (✅) or a red cross (❌). Below, we provide a detailed analysis of the highlighted errors:
>
> - **Floor 11**: Designated for art gallery use, Floor 11 is a sophisticated and flexible space designed
> to celebrate visual arts..... (✅).
> - .....
> - **Floor 32**: Floor 32 serves dual purposes, housing a renowned photography studio and corporate
> offices. ..... (❌).
> - .....
> - **Floor 34**: Transitioning into a leisure space, Floor 34 hosts a small cinema, providing an
> exclusive entertainment venue within the skyscraper (❌).
> - .....
> - **Floor 60**: This floor houses a luxury watch and timepiece atelier, celebrating the art of horology
> and fine craftsmanship. ..... (❌).
> - .....
> - **Floor 90**: Floor 90 offers a dynamic e-commerce and digital marketing center focused on online
> business innovation and consumer engagement strategies. ..... (❌).
>
>
> **Analysis of Errors**
>
> These examples reveal two common failure modes:
>
> - **Inconsistent Floor Allocation**: Some floors, such as 32 and 34, were incorrectly used for purposes outside the defined range, highlighting the model’s challenges in adhering strictly to range-based instructions.
>
> - **Unplanned Floor Use**: Floors like 60 and 90 were assigned purposes not outlined in the original instructions, suggesting that the model struggles to maintain instruction adherence over extended sequences.
>
> These findings mark an important contribution as LongGenBench is the first systematic study to examine long-form generation capabilities in extended contexts with a focus on instruction adherence. While some results may not appear surprising in isolation, our benchmark provides a rigorous, comprehensive framework that reveals consistent, quantifiable patterns in model behavior over long sequences. By identifying specific error cases, such as inconsistent allocation and unplanned floor use, we offer concrete insights into the limitations of current long-context LLMs. This structured approach enables the community to better understand and address the inherent challenges in long-form generation, laying the groundwork for future improvements in model architecture and training.

---

> ### Author Response · Authors · 2024-11-18
> **Comment (2/3)**
>
> **W2/Q2: Prompt Format and Complexity**
>
> Thank you for your suggestion and insights regarding prompt format and its influence on model output. We agree that prompt design can significantly impact model performance, as shown in prior research. To address this, we experimented with different prompt formats, including reordering structures, across several models, including LongWriter, Mistral, and Qwen 2. The results are summarized in the table below:
>
> | Prompt Format | Model               | CR   | STIC-2  | Length (word) | wAvg | Rank |
> |---------------|---------------------|------|----------|---------------|------|------|
> | Prompt - 1    | LongWriter-llama3.1-8b_maxlen16000      | 46.0%  | 9.83%    | 11036          | 4.5  | 3    |
> |               | Qwen2-7B-Instruct_maxlen16000           | 60.0%  | 16.13%   | 5138           | 9.7  | 2    |
> |               | Mistral-7B-Instruct-v0.2_maxlen16000    | 81.8%  | 17.44%   | 7296           | 14.3 | 1    |
> | Prompt - 2    | LongWriter-llama3.1-8b_maxlen16000_prompt_format | 24.3% | 8.35% | 6189 | 2.0 | 3 |
> |               | Qwen2-7B-Instruct_maxlen16000_prompt_format | 57.3% | 16.34% | 4334 | 9.4 | 2 |
> |               | Mistral-7B-Instruct-v0.2_maxlen16000_prompt_format | 62.3% | 16.29% | 4750 | 10.2 | 1 |
>
> Our experiments indicate that while prompt format variations do impact model performance, the relative ranking of model results remains consistent, even with different formats. This suggests that prompt structure alone does not substantially alter overall model performance trends or relative comparisons across models.
>
> Determining the optimal prompt format for each model is beyond the primary scope of our study. Instead, our focus was on ensuring prompt consistency across all models to evaluate their performance differences fairly. By using the same baseline format for each model, we controlled for prompt-induced variability, allowing observed performance differences to reflect the inherent capabilities of each model rather than prompt design differences.
>
> To provide further insights, we included an analysis of prompt complexity in Appendix G. This analysis examines the impact of different prompt structures on model performance.
>
>
> **W3/Q4: Reasoning Tasks and Adding a Reasoning Axis**
>
> Thank you for your suggestion regarding reasoning tasks and the potential addition of a reasoning axis in our benchmark. We agree that reasoning, particularly over extended contexts, is a crucial area for assessing LLMs. In our current benchmark, we address some aspects of temporal reasoning through periodic instructions embedded in the prompts. For instance, instructions such as “run long distance every 7 days” require the model to accurately apply this recurring activity at specified intervals. These periodic directives test the model's basic understanding of temporal structures within long-form text generation and allow us to inspect generated content systematically. This setup facilitates an automated evaluation of temporal reasoning without requiring labor-intensive human review.
>
> While dimensions such as reason, coherence and factual accuracy are undeniably valuable, they become meaningful only after the model demonstrates a foundational ability to follow structured instructions. Similarly, we consider instruction adherence to be a foundational aspect of long-form generation, as models must first reliably follow specified prompts to ensure quality outputs.
> By focusing initially on instruction-following, we aim to lay a pragmatic and impactful foundation for long-form text evaluation.

---

> ### Author Response · Authors · 2024-11-18
> **Comment (3/3)**
>
> **W4/Q5: Evaluating SOTA Models and Architectural Diversity**
>
> Thank you for your suggestion. We have conducted additional experiments to evaluate Mamba’s performance under various settings and compared it with existing baseline models, focusing on both inference speed and accuracy. Our findings show that Mamba significantly enhances computational efficiency but fails to maintain accuracy comparable to other models.
> The results are summarized in the table below:
>
> | Model              | Claimed Length (token) | CR    | STIC-1 | STIC-2 | Length (word) | Inference Time |
> |--------------------|-------------------------|-------|--------|--------|---------------|----------------|
> | INMamba-2.8B-16K   | 2K                      | 11.3% | 23.8%  | 2.0%   | 902           | 80s            |
> | Mamba-2.8B-32K     | 2K                      | 5.6%  | 29.8%  | 1.6%   | 846           | 80s            |
> | LLama-3.1-8B-16K   | 128K                    | 93.5% | 23.4%  | 22.0%  | 8804          | 2499s          |
>
> A limitation of the current Mamba model is its maximum training sequence length of 2,000 tokens, which restricts its ability to handle longer text sequences effectively. This limitation directly impacts its performance in long-context tasks, as observed in our results. While recent efforts, such as **LongMamba** [1], aim to extend Mamba’s capabilities to longer contexts via training-free receptive field enlargement, the model parameters are not yet available as open-source. We intend to test LongMamba once it becomes publicly accessible.
>
> [1] LongMamba: Enhancing Mamba’s Long-Context Capabilities via Training-Free Receptive Field Enlargement. ICLR-2025 Submission.
>
>
> **Q3: Complex prompts and instructions might need manyshots**
>
> Thank you for this insightful question. We chose not to incorporate few-shot or many-shot examples in our benchmark for two main reasons:
>
> **Context Length Limitations**: Given that our outputs already approach or reach the model’s maximum long-context length, adding few-shot examples would significantly reduce the space available for evaluating long-form generation capabilities. This reduction would compromise our assessment of the model’s performance on extended outputs, as it would limit the effective long-context sequence length available for instruction adherence.
>
> **Realistic Evaluation Setting**: Zero-shot evaluation aligns better with practical, real-world applications of long-context models, where explicit examples may not always be available in deployment scenarios. Major long-context benchmarks, such as LongBench [1], RULER [2], and NeedleBench [3], also primarily use zero-shot settings. This approach ensures a fair comparison across models without relying on specific prompt design strategies that could introduce variability in performance, allowing us to focus on the intrinsic instruction-following capabilities of each model.
>
> We appreciate your question and hope this clarifies our reasoning. Please let us know if further elaboration is needed.
>
> References:
>
> [1] LongBench: A Bilingual, Multitask Benchmark for Long Context Understanding.
>
> [2] RULER: What's the Real Context Size of Your Long-Context Language Models?
>
> [3] NeedleBench: Can LLMs Do Retrieval and Reasoning in a 1 Million Token Context Window?

---

> > ### Comment · Reviewer_dQFM · 2024-11-21
> > **response to authors**
> >
> > Thank reviewers for the response, and I remain positive of the paper and raised the score to 8.

---

> > > ### Author Response · Authors · 2024-11-21
> > >
> > > Many thanks four reviewing our paper and raising the score! We greatly appreciated your constructive review, which improved our paper!

---

### Author Response · Authors · 2024-11-18
**Common response**

Dear Reviewers,

Thank you for your thoughtful and constructive feedback on our paper. We have carefully addressed your comments and suggestions, as detailed below, which have helped us enhance the clarity and quality of our work. We also appreciate the generally positive feedback and remarks we received.

We have introduced a novel benchmark specifically designed to rigorously assess the capability of LLMs to generate extended texts while adhering to complex instructions. Mastery of instruction-following is fundamental for long-form text generation, as it establishes the topic and boundaries of the content. Building on this foundation, extended texts must exhibit advanced qualities such as sustained logical reasoning over long contexts, narrative coherence, and originality in writing. Importantly, these advanced qualities can only be meaningfully evaluated once models demonstrate strong proficiency in following instructions for long-form text generation. As discussed in the "Analysis and Limitations" section, we openly acknowledge the current constraints of our study. We recognize that achieving effective long-context text generation is a complex and long-term challenge that demands sustained effort. Nonetheless, we believe our focus on evaluating extended texts through the lens of instruction-following capabilities offers a pragmatic and impactful starting point. This approach lays a solid foundation for addressing the broader challenges associated with long-context generation, providing valuable insights for the community.

We have addressed the concerns and suggestions from each reviewer as follows:

**Main Revision: (In our rebuttal version PDF)**

1. **Clarified Evaluation Process (Reviewers p7FN, hgz3, bHUs) - Appendix C**

We revised the paper to provide a clearer description of the evaluation process. This helps in illustrating the evaluation steps more intuitively.

2. **Clearer Distinction Between STIC-1 and STIC-2 (Reviewers p7FN, bHUs) - Appendix E**

We clarified the distinctions between STIC-1 and STIC-2 with examples in the appendix. This revision highlights how each component assesses different aspects of instruction adherence, including task complexity and sequential requirements.

3. **Added more different model Experiments to Table 3 (Reviewers dQFM, Nybu):**

We included additional experiments with models such as mamba-2.8B, phi-3-mini-128K-instruction, phi-3.5-MOE-128-instruction, and FILM-7B in Table 3. This broader evaluation provides a more comprehensive view of model performance across different architectures.

4. **Impact of Different Prompt Formats (Reviewer dQFM): - Appendix G**

To address the effect of prompt format on model performance, we added a new analysis in Appendix G that evaluates the impact of prompt structure on adherence and generation quality.

5. **Added Symbol Explanation Table in Appendix (Reviewer bHUs) - Appendix A**

To aid readers’ understanding, we included a table in Appendix A that explains all symbols used in the paper, ensuring consistency and clarity throughout.

6. **Some minor detail revisions in the main text (Reviewer bHUs)**

We made several minor revisions throughout the main text, including clarifying definitions of terms like “main task,” “subtask,” “instruction task,” and “specific task,” as well as defining “CR” (Completion Rate) in Table 3.

We believe these modifications have significantly improved the quality, clarity, and comprehensiveness of our work in the revised version. We thank you again for your valuable feedback and time, which helped us make these impactful improvements.


Best regards,

Authors of Submission Number: 3588

---

### Author Response · Authors · 2024-11-21

Dear Reviewers,

Thank you once again for your valuable feedback and time reviewing our paper.

We have carefully addressed your comments and suggestions, implementing several revisions that we believe significantly enhance the clarity, depth, and contributions of our work. Your constructive feedback has been instrumental in these improvements.

We would like to kindly encourage you to participate in the ongoing discussion phase. We are eager to address any further questions or concerns you may have and provide additional clarifications or evidence to support our work. Your insights are invaluable to refining our research and ensuring its relevance and impact.

Additionally, we hope that the changes and clarifications we made can prompt a reconsideration of your evaluation, as we believe these updates align closely with the constructive suggestions provided.

Please do not hesitate to reach out with any remaining concerns or queries. We are committed to ensuring all aspects of our submission are adequately addressed.

Many thanks!


Best regards,

Authors

---

### Author Response · Authors · 2024-11-22
**Common Response - 2**

Dear Reviewers,

Thank you once again for taking the time to review our paper and for your valuable and constructive feedback. We deeply appreciate your thoughtful insights, which have been instrumental in enhancing the clarity, depth, and overall quality of our work.

We would like to extend our special thanks to Reviewers dQFM and bHUs for following up on our responses. Your engagement has been particularly helpful. We also encourage the remaining reviewers to review our responses and share any additional questions or clarifications. We remain happy to address them promptly.

To facilitate your review, we have highlighted all revisions in red throughout the manuscript. The primary additions, including expanded discussions and supplementary details, are located in the appendix for your convenience.

We sincerely value your efforts and insights in this process and look forward to any further feedback you may have.

Best regards,

Authors of Submission Number: 3588

---

### Meta-Review · Area_Chair_7MQd · 2024-12-17

**Metareview:**

This paper proposes LongGenBench, a novel benchmark to evaluate the skills of LLMs in generating texts with an average of 20K. These generation tasks are synthetically constructed for two domains, diary/menu writing (e.g., plan menus for each week of the year) and building design/urban planning (design a 100-floor skyscraper). Evaluation is performed using LLMs-as-judge, where a long generated text is decomposed into individual components (e.g., design plan for each floor of the skyscraper), and each component is rated using an LLMs for correctness. Evaluation revealed that most state-of-the-art LLMs struggle with generating coherent texts beyond 10K+ tokens, despite registering strong results on existing long context benchmarks, such as Needle-in-a-Haystack and Ruler.

Most reviewers recognized the novelty and significance of this paper as the first benchmark on measuring LLM performance in generating extremely long texts (dQFM, p7FN, hgz3, bHUs). The benchmark is also designed in a clever way to facilitate automated evaluation and “enable reliable and accurate evaluation for long-form capability” (bHUs). The benchmark “uncovers a setting that seems to be challenging to SOTA models” (hgz3). The paper is also “well written” (Nybu) and “easy to follow” (dQFM, p7FN).

The major weaknesses are raised by Reviewer hgz3 and Nybu, namely the limited scope of synthetic domains and scenarios (hgz3, Nybu), as well as the comprehensiveness of the evaluation metrics (Nybu), such as being too task-specific and unable to capture more interesting aspects such as creativity. There are also other issues raised in the review, such as potential correctness issues using LLM-as-judge, but they are addressed during the rebuttal phase with additional experiment results.

While there are still open questions around the limitations of using synthetic domains and improving evaluation methodologies using deterministic metrics besides LLMs-as-judge, as the research direction of long text generation / CoT is receiving more traction, we believe that this paper serves a timely and significant contribution to the community by offering the first benchmark in this area. The evaluation method is also already a step forward compared to classical Needle-in-the-Haystack approaches. Therefore, the decision is Acceptance.

**Additional Comments On Reviewer Discussion:**

Please refer to the metareview.

---

### Decision · Program_Chairs · 2025-01-22

Accept (Poster)